# Structural dynamics determine voltage and pH gating in human voltage-gated proton channel

Shuo Han, Sophia Peng, Joshua Vance, Kimberly Tran, Nhu Do, Nauy Bui, Zhenhua Gui, Shizhen Wang*

Department of Cell Biology and Biophysics, School of Biological and Chemical Sciences, University of Missouri-Kansas City, Kansas City, United States

**Abstract** Voltage-gated proton (Hv) channels are standalone voltage sensors without separate ion conductive pores. They are gated by both voltage and transmembrane proton gradient (i.e., ΔpH), serving as acid extruders in most cells. Like the canonical voltage sensors, Hv channels are a bundle of four helices (named S1 –S4), with the S4 segment carrying three positively charged Arg residues. Extensive structural and electrophysiological studies on voltage-gated ion channels, in general, agree on an outwards movement of the S4 segment upon activating voltage, but the real-time conformational transitions are still unattainable. With purified human voltage-gated proton (hHv1) channels reconstituted in liposomes, we have examined its conformational dynamics, including the S4 segment at different voltage and pHs using single-molecule fluorescence resonance energy transfer (smFRET). Here, we provide the first glimpse of real-time conformational trajectories of the hHv1 voltage sensor and show that both voltage and pH gradient shift the conformational dynamics of the S4 segment to control channel gating. Our results indicate that the S4 segment transits among three major conformational states and only the transitions between the inward and outward conformations are highly dependent on voltage and pH. Altogether, we propose a kinetic model that explains the mechanisms underlying voltage and pH gating in Hv channels, which may also serve as a general framework for understanding the voltage sensing and gating in other voltage-gated ion channels.

*For correspondence:
wangshizhen@umkc.edu

## Editor's evaluation

This study examines the influence of voltage on conformational dynamics of voltage-sensing Hv1 channel at a single molecule resolution. Previously it was shown that although Hv1 channels lack a separate pore domain unlike most members of the voltage-gated channel family, the pore opening and voltage-sensing are distinct but linked processes. This study provides new insight in the mechanism of gating by showing that the voltage-sensor is able to access an intermediate conformation distinct from the activated and resting state.

## Introduction

Voltage-gated cation channels mediate the electrical excitability of many cells, such as cardiomyocytes, muscle, and nerve cells. As a unique subfamily, voltage-gated proton (Hvs) channels are solely composed of voltage sensors without the separate pore-forming domains typically found in canonical voltage-gated cation channels (*Ramsey et al., 2006*; *Sasaki et al., 2006*). Hv channel voltage sensors are hourglass shaped with a bundle of four helices (named S1–S4), like those of the voltage-gated cation channels (*Takeshita et al., 2014*; *Bayrhuber et al., 2019*). The S4 segment of the

voltage-gated cation channels is highly conserved and contains multiple positively charged residues to sense membrane voltage (*Gonzalez et al., 2010*). High resolution structures of the voltage sensors from different organisms, including Hv channels, largely agree on a model where the charge carrying S4 segment is driven outward by depolarization voltage – a movement that is then transduced to gate the channel pore (*Guo et al., 2016*). Despite many years of research, the conformational trajectories of the voltage sensor, as well as the underlying mechanisms by which voltage and pH drive these conformational transitions in Hv channels, remain unclear and, sometimes even controversial. For example, accessibility assays showed that only the middle Arg in the S4 segment of the Hv channel switches its accessibility upon voltage activation (*Gonzalez et al., 2013*), supported by the crystal structure of a voltage-sensitive phosphatase and the electron paramagnetic resonance data of the human Hv channel (hHv1) (*Li et al., 2015*; *Li et al., 2014*). However, the gating currents of Hv channels and many other voltage-gated cation channels suggested approximately three Arg residues move across the membrane barrier upon gating transition (*Carmona et al., 2018*; *De La Rosa and Ramsey, 2018*). Furthermore, a unique function-defining feature of Hv channels is that their voltage depen-dence is shifted by proton concentrations at the intra- and extracellular sides (i.e., transmembrane pH gradient) (*Cherny et al., 1995*). As a result, Hv channels are only activated when the proton electrochemical gradient is directed outwards (*DeCoursey, 2013*). However, it is still unclear how pH modulates the voltage sensitivity of the Hv channels. Titratable residues were proposed to be involved in the pH regulation of Hv channels and H168 has been identified as a critical intracellular pH sensor in hHv1 channels (*Cherny et al., 2015*; *Cherny et al., 2018*). In addition, the most recent studies indicated that both pH gradient and voltage alter the dynamics of the Hv channel gating currents that reflect the movements of the S4 segment (*DeCoursey, 2018a*; *Carmona, 2021*).

To understand the structural basis underlying voltage and pH gating in the hHv1 channel, we utilized the single-molecule fluorescence resonance energy transfer (smFRET) approach that directly visualized the real-time conformational transitions and dynamics of hHv1 channels in lipid environ-ments. In the present work, smFRET measurements were performed at the voltages and pH gradients that the gating status of hHv1 channels has been unambiguously defined. Our results indicate that both voltage and pH gradient modify the conformational dynamics of the S4 segment in hHv1 chan-nels. Our studies provide not only a mechanistic rationale for the voltage–pH interplay in determining hHv1 channel gating but also structural insights into voltage sensing in other voltage-gated cation channels.

## Results
### Conformational dynamics of the hHv1 channels driven by voltages

The codon-optimized hHv1 WT channels with an N-terminal 6× histidine tag were expressed and purified from *E. coli* host cells, exhibiting a single, symmetric peak in size-exclusion chromatography profile (*Figure 1—figure supplement 1A*). The hHv1 WT protein was reconstituted into liposomes (POPE/POPG = 3/1, wt/wt, protein/lipid = 1/200, wt/wt) and its channel function was examined by liposome flux assay (*Figure 1—figure supplement 1A*). Our results indicated that the purified hHv1 WT protein retained its native proton channel function (*Figure 1—figure supplement 1C*).

To implement smFRET studies on the hHv1 channel, we mutated the two intrinsic Cys107 and Cys249 residues into Ser, then introduced cysteine mutations, one pair at a time, at different sites of the hHv1 channel. The goal was to uncover the relative movements among the four transmem-brane segments at different pHs/voltages by measuring the real-time FRET changes between the donor and acceptor fluorophores conjugated with the cysteine residues which were introduced at different sites. These movements reported by FRET were mapped onto the available Hv channel structures (*Takeshita et al., 2014*; *Bayrhuber et al., 2019*; *Geragotelis et al., 2020*) to reconstruct the conformational transitions and their dynamic behaviors (*Figure 1A*). The hHv1 channel mutants were expressed, purified, and then labeled by the chemically modified Cy3/Cy5 FRET fluorophore pair with improved photostabilities (*Zheng et al., 2017*). The fluorophore-labeled hHv1 channel proteins were reconstituted into liposomes (POPE/POPG = 3/1, wt/wt, 5 mg/ml) with an extremely low protein lipid ratio of 1:4000 (wt/wt), so that most liposomes were either empty or only contained one hHv1 channel. Moreover, hHv1 channels in liposomes exist mainly as monomers because the concentration of hHv1 proteins during reconstitution is only 38 nM, while the dissociation constant of homodimeric

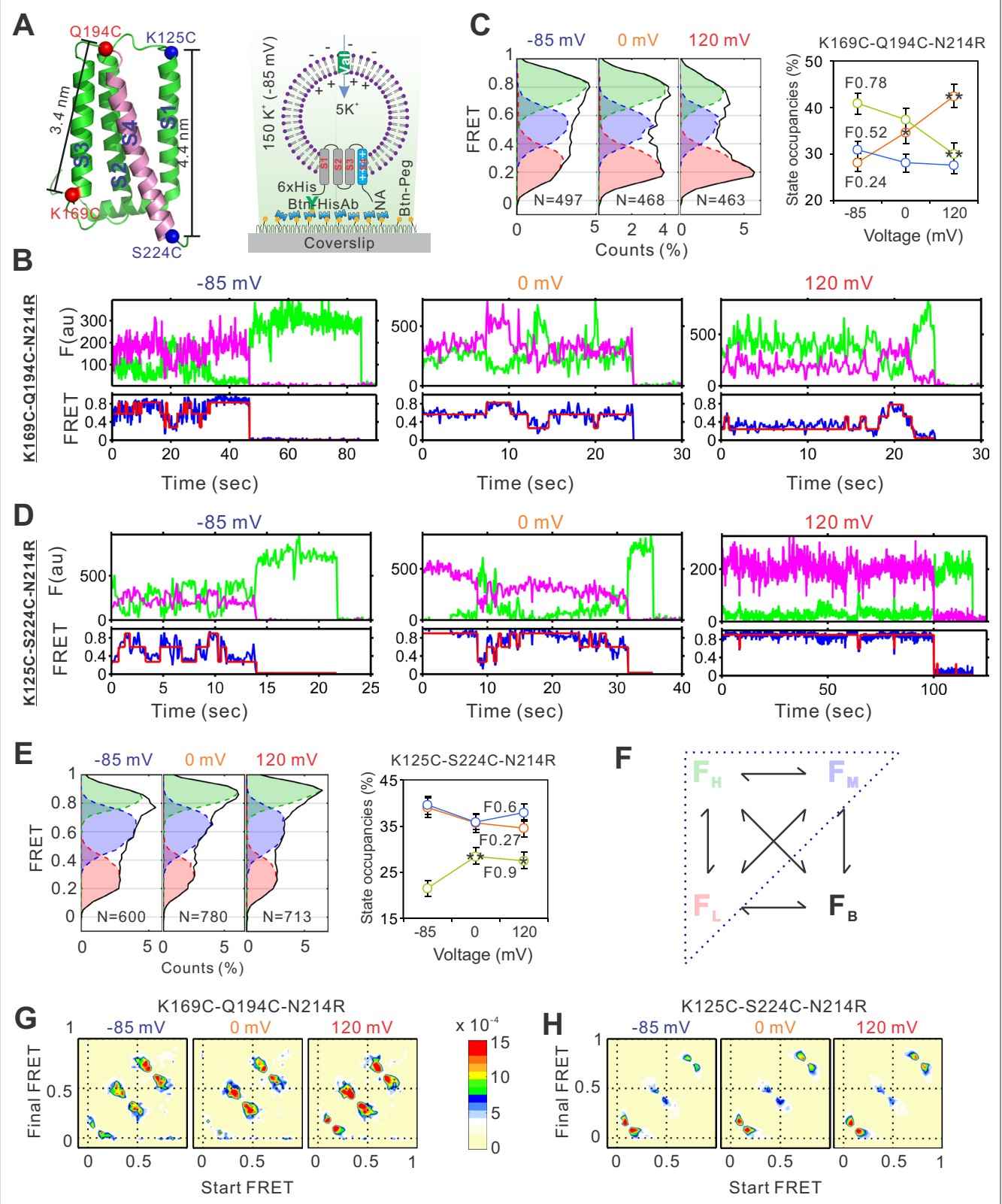

**Figure 1.** The voltage-dependent conformation dynamics of the hHv1 channel S4 segment revealed by smFRET. (**A**) Left panel, the cartoon of the hHv1 NMR structure (5oqk). The K125-S224 (blue) and K169-Q194 (red) labeling sites were highlighted by spheres; right panel, the sample immobilization configuration for smFRET imaging. The PEG passivated surface with 2% PEG-biotin (Btn-Peg) binds the biotinylated anti-Histag antibodies (Btn-HisAb) via neutravidin (NA), then the hHv1 liposomes with N-terminal 6*Histag (6×His) facing the outside were selectively retained for smFRET imaging. The

*Figure 1 continued*

liposome voltages were controlled by the K$^+$ gradient across liposomes in the presence of K$^+$ ionophore valinomycin (Val). Representative smFRET traces between the K169C-Q194C (**B**) and K125C-224C (**D**) labeling sites under −85 mV (resting), 0 mV (weak), and 120 mV (strong activating) voltages. The green and pink lines are donor and acceptor fluorescence intensities; the blue and red lines are the real and idealized FRET. FRET histograms and state occupancies of the smFRET data from the K169C-Q194C (**C**) and K125C-224C (**E**) labeling sites under different voltages. State occupancy data presented as mean ± SE, *N* is the number of smFRET traces. Unpaired *t*-tests were performed to examine significance levels of voltage-induced FRET state occupancy changes, in comparison to the occupancy of the same FRET state at −85 mV, with * and ** indicating p < 0.05 and p < 0.01. (F) The structure model used for kinetic analysis of smFRET traces, contains four FRET states, including low ($F_L$), medium ($F_M$), high ($F_H$), and bleaching/blinking states ($F_B$). However, the occupancies of the $F_B$ state are extremely low and the transitions to the $F_B$ state are very rare at all labeling sites and voltage/pH conditions, therefore the $F_B$ state was ignored in the following kinetic analyses. Transition density plots from the smFRET data at the K169C-Q194C (**G**) and K125C-224C (**H**) labeling sites under different voltages.

The online version of this article includes the following figure supplement(s) for figure 1:

**Figure supplement 1.** The purified hHv1 protein functions as a proton channel.

**Figure supplement 2.** The N214R mutation abolishes hHv1 channel proton inhibition by preventing proton uptake into the liposomes.

**Figure supplement 3.** The voltage-induced conformational changes of the hHv1 channels revealed by single-molecule fluorescence resonance energy transfer (smFRET).

**Figure supplement 4.** Representative single-molecule fluorescence resonance energy transfer (smFRET) traces of the K169C-Q194C.

hHv1 proteins is ~3.2 µM (*Li et al., 2015*). The orientation of the hHv1 channel in proteoliposomes was controlled by incubating the samples on the PEG passivated coverslip surface coating with anti-Histag antibodies (MA1-21315-BTIN, Invitrogen), so that only those with the N-terminal His-tag of the hHv1 facing outside were retained for smFRET imaging. Thus, the defined electrical potential was generated with K$^+$ gradients and K$^+$ ionophore valinomycin and applied to the hHv1 channel in liposomes (*Figure 1A*). The smFRET imaging was performed on a customized TIRF microscope with a dual-view beam splitter and EMCCD camera as described previously (*Wang et al., 2019*; *Wang et al., 2016*), and the raw movies collected without any corrections were processed by the SPARTAN software to identify individual molecules, extract donor/acceptor intensities, and calculate FRET (*Juette et al., 2016*). The liposome voltages were calculated with the Nernst equation and switched by changing the extraliposomal K$^+$ concentrations (*Figure 1A*). Our initial smFRET measurements did not detect any significant voltage-dependent FRET changes at the voltage-sensing S4 segment (*Figure 1—figure supplement 2A*). We then realized that the proton uptake through the hHv1 channel will decrease the extracellular pH of hHv1 channels (i.e., the intraliposomal pH) and start to deactivate them in less than 1 min (*Figure 1—figure supplement 2B*). As a result, the duration of the activated conformations was too short to be captured. To circumvent this problem, we introduced the N214R mutation, which was shown to significantly reduce the outward H$^+$ currents without altering the voltage dependence of hHv1 channels (*Carmona et al., 2018*; *De La Rosa and Ramsey, 2018*). We confirmed that hHv1 N214R proteins reconstituted into liposomes exhibit no detectable proton uptake within 30 min (*Figure 1—figure supplement 2B*, upper left panel). The hHv1 channel proteins for smFRET studies contain multiple mutations with attached fluorophores. Our liposome flux assay data indicated that, despite these modifications, most hHv1 proteins without the N214R mutation remained functional to mediate proton uptake (*Figure 1—figure supplement 2B, C*). On the N214R background, substantial voltage-dependent FRET changes were detected at both the K169C-Q194C and K125C-S224C labeling sites, which exhibited reverse FRET changes upon activating voltages (*Figure 1A*, *Figure 1—figure supplement 3A*). SmFRET traces from the K169-Q194 labeling sites, exhibited large and spontaneous transitions among three major FRET states with centers at 0.24, 0.52, and 0.78, respectively, at both the resting (−85 mV) and activating voltages (*Figure 1B*, *Figure 1—figure supplement 4A*). For all these traces, the donor and acceptor fluorescence intensities were strongly anticorrelated, indicating that the FRET changes truly resulted from the structural changes rather than fluorophore photophysics (*Figure 1B*, *Figure 1—figure supplement 4A*). We fitted the histogram from the smFRET data collected at all voltages with one to five FRET states and the SSE (sum of squares due to errors) of Gaussian fits decreases >95% with three FRET states (*Figure 1—figure supplement 3B*). Therefore, we performed kinetic analysis using a model containing low ($F_L$), medium ($F_M$), and high ($F_H$) FRET states. We also introduced the $F_B$ FRET state for some blinking or bleaching events that the SPARTAN software occasionally failed to exclude (*Figure 1F*). However, the occupancies of the $F_B$ state were extremely low and transitions to the $F_B$

state were rare, therefore they were ignored in the kinetic analyses follow-up. The smFRET traces were then idealized using the Maximum Point Likelihood (MPL) algorithm (*Qin et al., 2000*), which allowed us to fix the centers of FRET states and analyze multiple datasets collected at different conditions. The FRET state occupancies calculated from several hundreds of hHv1 channels at different voltages showed that the weak (0 mV) and strong activating (120 mV) voltages significantly enriched the low FRET-0.24 state, while diminishing the medium and high FRET state (*Figure 1C*). Shifts of the FRET state occupancies indicated that activating voltage altered the conformational distributions of the S4 segment by enriching those at the extracellular side. The results were further validated by the smFRET data collected from the K125C-S224C sites (*Figure 1A*), which also exhibited large, spontaneous FRET transitions among 3 FRET states with centers at 0.27, 0.6 and 0.9 (*Figure 1D*, *Figure 1—figure supplement 4B*). The activating voltages (0 and 120 mV), in sharp contrast, promoted the high FRET 0.9 state at the K125CS224C labeling sites (*Figure 1E*). The voltage-dependent changes in FRET state occupancies were also reflected in the transition density plots, which showed that activating voltages promoted the transitions between the low FRET state at the K169C-Q194C sites and the high FRET state at the K125C-S224C sites (*Figure 1G, H*). Our smFRET data, from both the K169C-Q194C and K125CS224C labeling sites, consistently indicated that the S4 segment of hHv1 undergoes spontaneous transitions at equilibrium state under all voltages, and the activating voltage stabilizes the S4 segment at the extracellular side.

## Global conformational changes of the hHv1 channels at resting and activated voltages

To uncover the global conformational changes of hHv1 channels, we systematically examined the FRET distributions at multiple labeling sites, one pair at a time, at different voltages. The smFRET traces of all these labeling sites were analyzed with the 3 + 1 FRET state model using the Maximum Point Likelihood (MPL) algorithm (*Figure 1F*), except for the K125C-K169C sites, which fit better with a 2 + 1 FRET state model. Among the labeling sites we examined, FRET state occupancies of the S98C-Q194C sites, which reflect the movement of S1 vs S4, demonstrated similar voltage-dependent changes as that of K169C-Q194C sites, although not statistically significant (*Figure 2*). However, at all other sites including the Q194C-S219C (S4 itself), S98C-N132C (S1 vs S2), and K125C-K169C (S1 vs S3), FRET state occupancies did not exhibit any significant shifts that could be potentially associated with voltage gating (*Figure 2*). These data suggested that voltage primarily acts on the voltage-sensing S4 segment to promote its outward conformations, while the S1, S2, and S3 segments probably do not undergo significant relative movements between each other during gating transition.

Our studies were performed on the hHv1 channels at equilibrium state, as shown by the FRET contour maps, the FRET distributions did not change over time (*Figure 1—figure supplement 3A*). However, the S4 segment appeared to be highly dynamic and can sample all conformational states, at all voltages from −85 to 120 mV. These structural transitions potentially underlie the stochastic gating transitions found in voltage-gated ion channels. At a strong resting voltage of −85 mV, hHv1 channels should mostly be stabilized at the closed state. At a strong activating voltage of 120 mV, however, hHv1 channels may not necessarily reach the full opening state. A previous electrophysiological study indicated that the maximum open probability recorded from hHv1 channels in human eosinophils, even at the optimal pH gradient, is ~only 0.6 (*Cherny et al., 2003*). Therefore, incomplete gating transitions may lead to the relatively small changes in FRET state occupancies observed in our studies.

## pH gates hHv1 channels by modifying the dynamics of the S4 segment

To understand how pH shifts the voltage sensitivity of the hHv1 channel, smFRET measurements were performed on the K125C-S224C and K169C-Q194C sites at different pHs (*Figure 3A*). At 0 mV, lowering the intracellular pH (i.e., the extraliposomal pH) to 6.5 remarkably enhanced the high FRET 0.9 state and decreased the low FRET 0.27 state occupancies (*Figure 3A*, left). Reverse changes in FRET occupancies were observed at the K169C-Q194C labeling sites, where the low FRET state (F0.24) increased and the high FRET state (F0.78) decreased upon intracellular pH 6.5 (*Figure 3B*, left panels). FRET occupancy data from both labeling sites support that lowering the intracellular pH enriches the outward conformations of the S4 segment, consistent with its effect to shift the voltage dependence of hHv1 channels negatively, therefore promoting openings (*Cherny et al., 1995*). In sharp contrast, changing both the intra- and extracellular pHs symmetrically only shifted the FRET

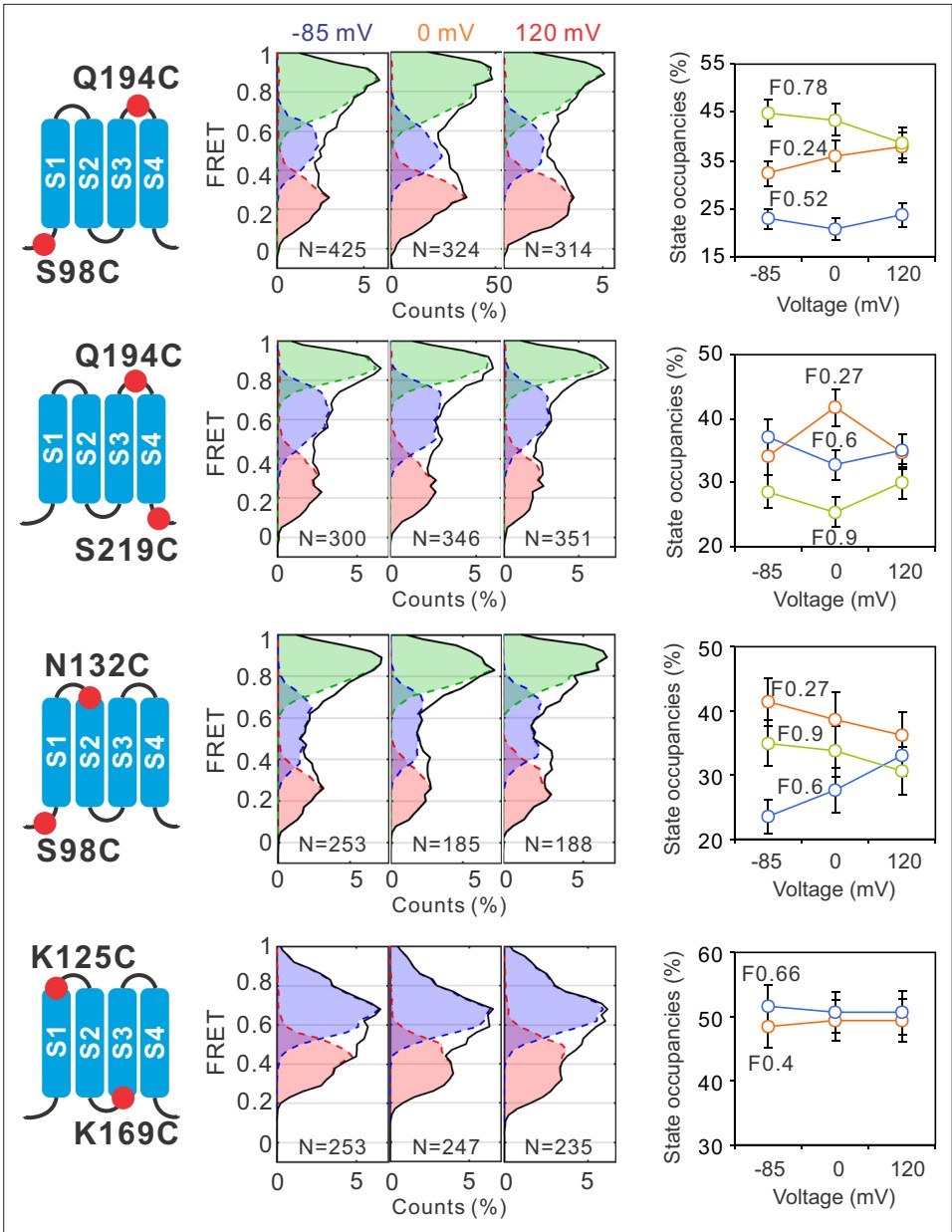

**Figure 2.** Global conformational changes of hHv1 channels driven by voltages. FRET histograms and state occupancies were calculated from the single-molecule fluorescence resonance energy transfer (smFRET) data at the S98C-Q194C, Q194C-S219C, S98C-N132C, and K125C-K169C labe−ling sites on the N214R background. All data were analyzed with the four state kinetic model as shown in *Figure 1F*, except for the K125C-K169C sites with a three state kinetic model containing F0.66, F0.4, and the $F_B$ state for bleaching/blinking events. The state occupancy data presented as mean ± SE, *N* is the number of smFRET traces. Unpaired *t*-tests were performed to examine the significance levels of voltage-induced FRET state occupancy changes, in comparison to the occupancy of the same FRET state at −85 mV, but none of these changes reached the significant level of $p < 0.05$.

occupancies slightly (*Figure 3A, B*, middle panels), agreeing well with the electrophysiological results that show the transmembrane pH gradient shifts the voltage dependence of hHv1 channels (*Cherny et al., 1995*). To further confirm our results, we introduced the H168Q background mutation, which was previously reported to attenuate the sensitivity of hHv1 channels to intracellular pH (*Cherny et al., 2018*). The results indicated that FRET distributions at both labeling sites no longer responded to intracellular pH changes (*Figure 3A, B*, right panels). Our data showed that pH gating of the hHv1 channels originates from its effect on the conformational dynamics of the S4 segment.

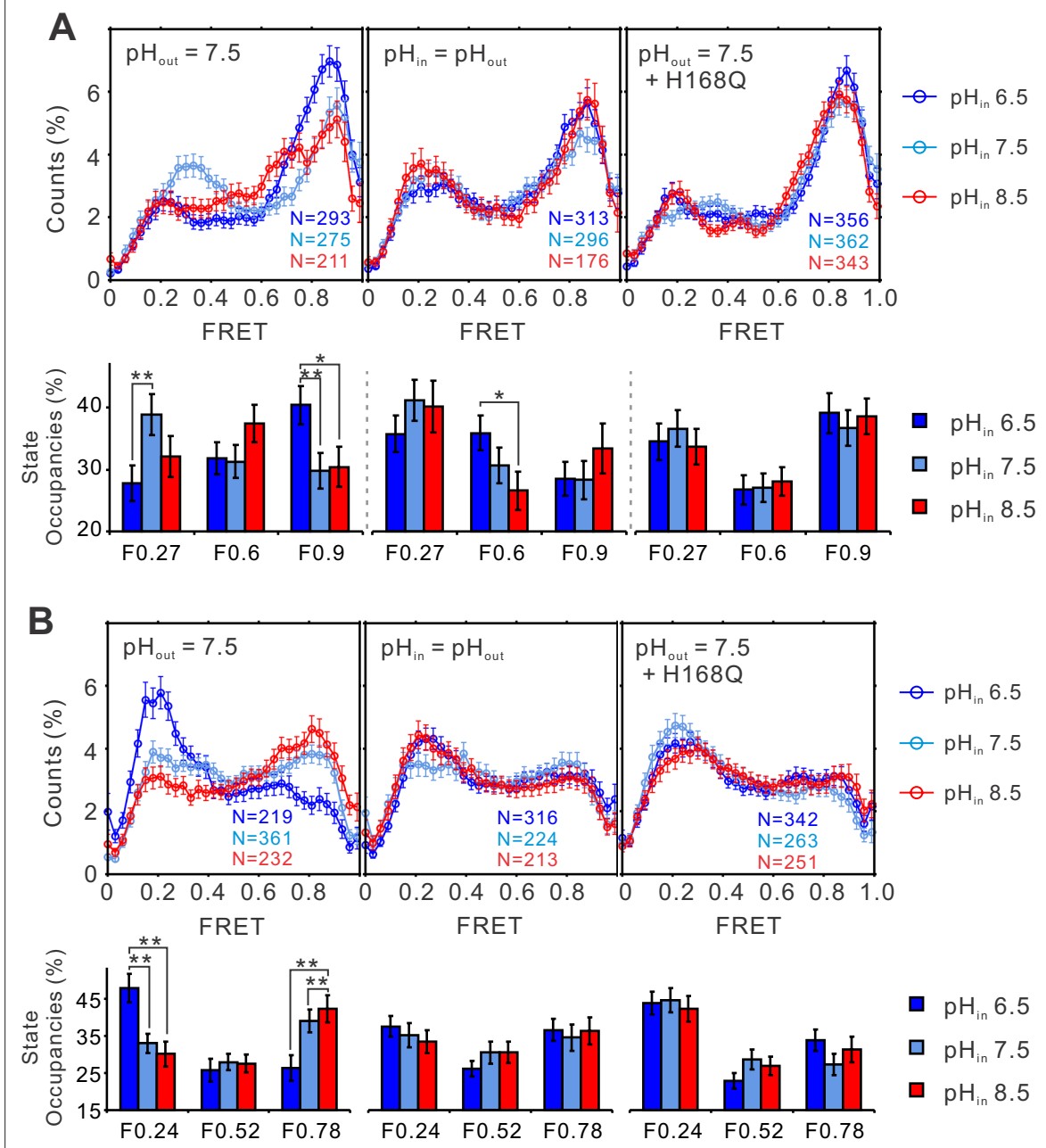

**Figure 3.** The structural dynamics of the S4 segment are pH dependent. FRET histograms and state occupancies of the single-molecule fluorescence resonance energy transfer (smFRET) data from the K125C-S224C (**A**), K169C-Q194C (**B**) of the hHv1 channels on the N214R mutation background, in liposomes under 0 mV with asymmetrical (left and right panels), symmetrical (middle) pHs on the WT (left, middle panels), or H168Q mutation backgrounds. State occupancy data presented as mean ± SE, $N$ is the number of smFRET traces. Unpaired $t$-tests were performed to examine the significance levels of pH-induced FRET state occupancy changes, in comparison to the occupancy of the same FRET state at pH 6.5, with * and ** indicating $p < 0.05$ and $p < 0.01$.

To elucidate the pH–voltage interplay in determining hHv1 channel gating, we examined the changes in FRET distributions induced by voltages at intracellular pH 8.5, which shifts the voltage dependence of hHv1 channels positively. Consistently, voltage-dependent enrichments of the high FRET 0.9 state at the K125C-SS24C sites were almost abolished (**Figure 4A**, right and middle panels). The potential of activating voltage in enriching the low FRET 0.24 state was also significantly attenuated at the K169C-Q194C sites (**Figure 4B**, right and middle panels). As shown in **Figure 3**, the H168Q mutation not only abolished the dependence of FRET distributions on intracellular pH but

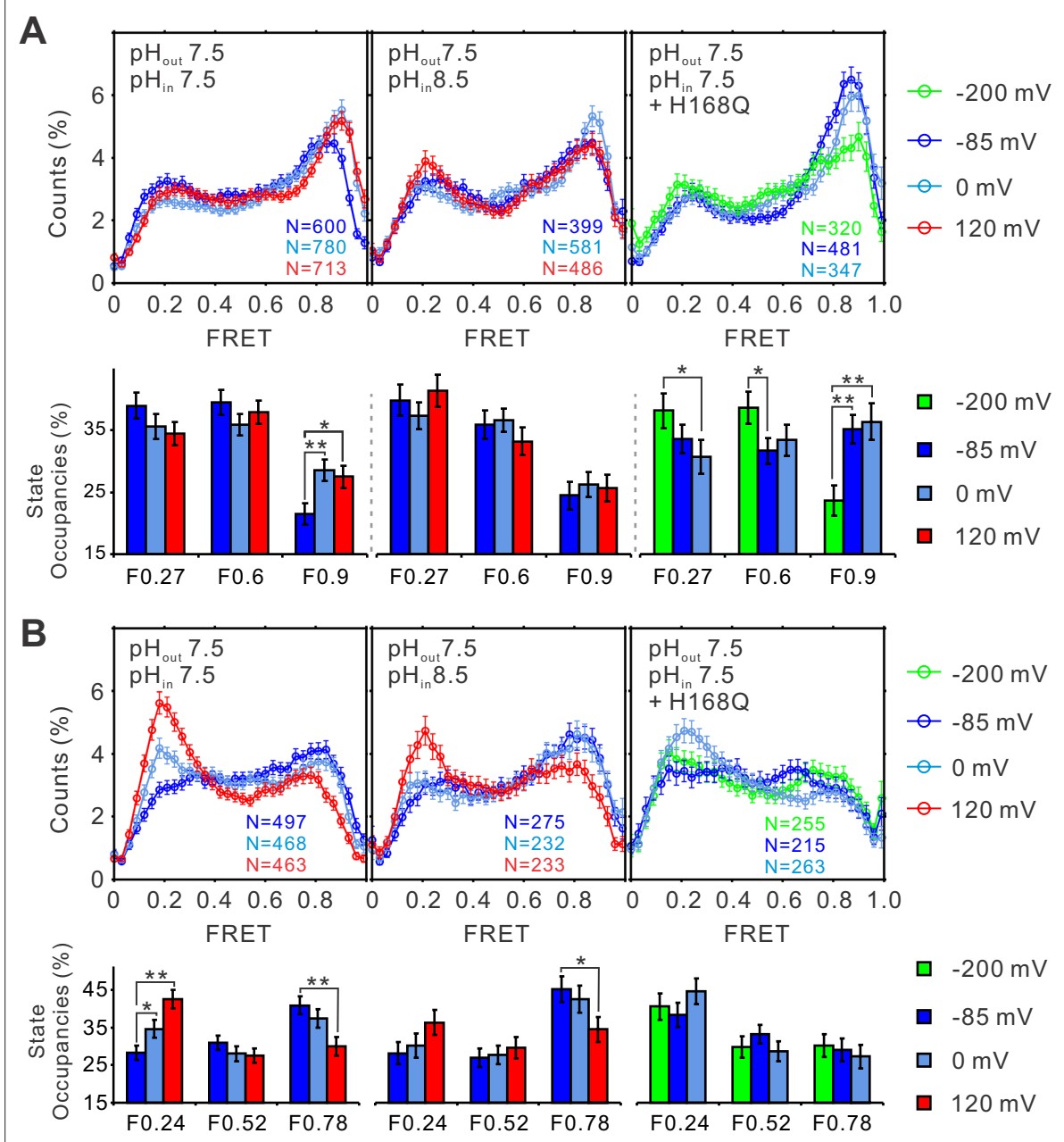

**Figure 4.** The structural dynamics of the S4 segment are determined by voltage–pH interplay. FRET histograms and state occupancies of the single-molecule fluorescence resonance energy transfer (smFRET) data from the K125C-S224C (**A**) and K169C-Q194C (**B**) labeling sites of the hHv1 channels with N214R mutation at different voltage/pH conditions on the WT (left, middle panels) and H168Q mutation backgrounds (right panels). State occupancy data presented as mean ± SE, $N$ is the number of smFRET traces. Unpaired $t$-tests were performed to examine the significance levels of voltage-induced FRET state occupancy changes, in comparison to the occupancy of the same FRET state at −85 or −200 mV on the H168Q mutation background, with * and ** indicating $p < 0.05$ and $p < 0.01$.

also enriched the high FRET 0.9 state at the K125C-S224C sites and the low FRET 0.24 state at the K169C labeling sites, at all intracellular pHs tested. These data propose that the H168Q mutation may stabilize the outward conformations of the S4 segment, therefore more negative voltage may be required to drive the S4 segment toward the resting state. Indeed, we found that the enrichment of the low FRET 0.27 populations correlating with channel closure was only observed on the K125C-S224C labeling sites at a more negative voltage (−200 mV) on the H168Q background, unlike those on the WT background that can be observed on the less negative voltage of −85 mV (***Figure 4A***,

left and right panels). These data suggest that the H168 residue may be involved in sensing intracellular pH through direct or indirect interactions with the S4 segment by stabilizing its inward resting conformations. However, only very minor effects were observed on the K169C-Q194C labeling site, perhaps due to the local impacts by the K169C mutations and conjugated fluorophores (*Figure 4B*, right panel).

## The structural transitions are dependent on the voltage/pH

To provide mechanistic insights into the voltage and pH gating in hHv1 channels, we performed kinetic analysis using the 3 + 1 state model as described previously with the MPL algorithm (*Figure 1F*). The smFRET traces at different voltage/pH conditions were idealized and the rate constants of transitions among the low, medium, and FRET states were obtained. We then calculated the equilibrium constants ($K_{eq}$) of different transitions as ratios of the forward ($K_F$) to reverse ($K_R$) transition rates. At

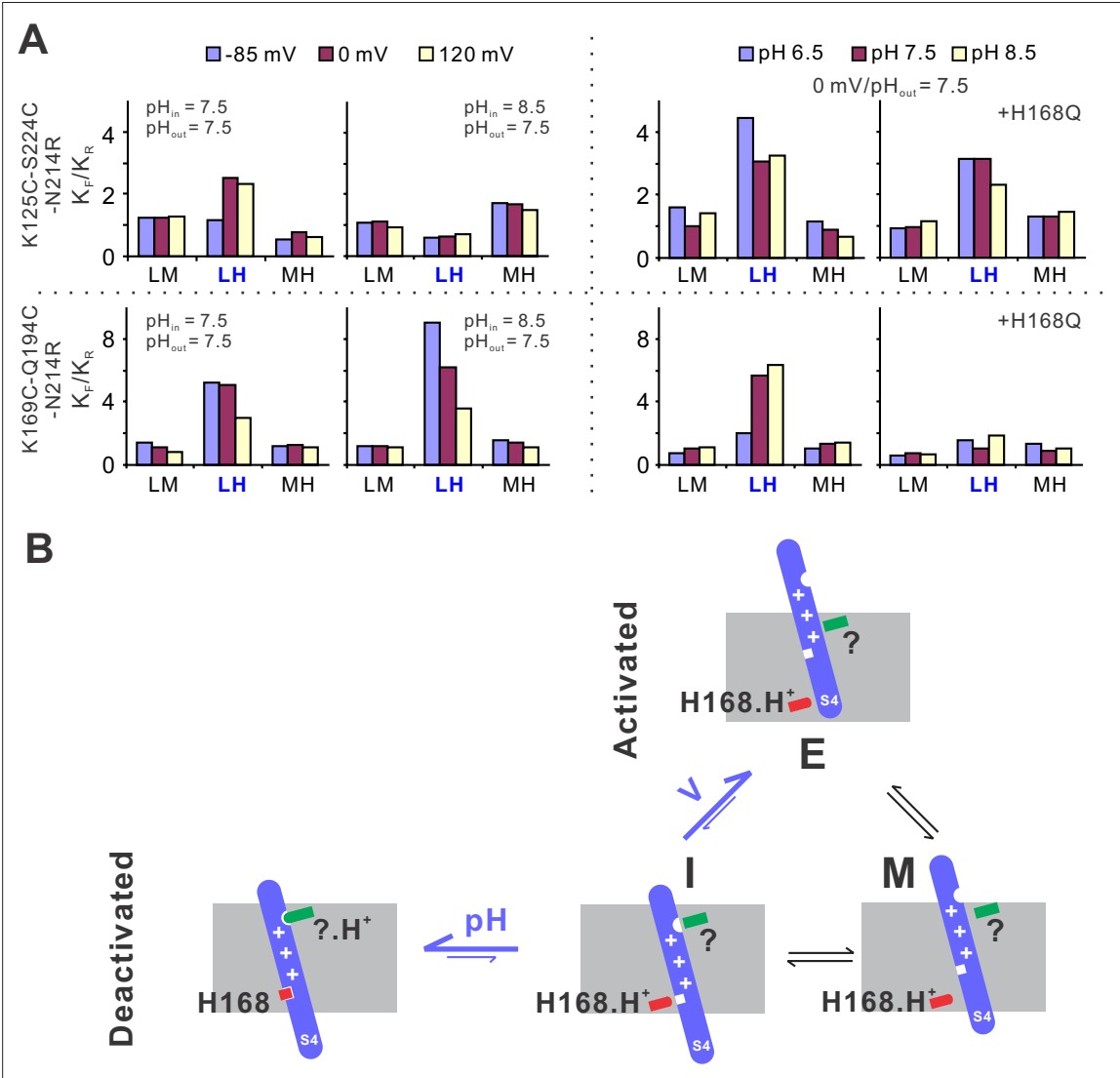

**Figure 5.** The structural basis underlying voltage–pH gating of hHv1 channels. (**A**) Voltage- and pH-dependent changes in equilibrium constants ($K_{eq}$), as the ratio of forward ($K_F$) and reverse ($K_R$) transition rates. The transition rates were calculated from the single-molecule fluorescence resonance energy transfer (smFRET) traces at different voltage/pH conditions idealized using the Maximum Point Likelihood (MPL) algorithm with the 4 FRET state kinetic model. (**B**) The structural model to explain the voltage and pH gating in hHv1 channels. The S4 segment of hHv1 channels has three distinct conformations according to their positions in membranes, that is the intracellular (I), middle (M), and extracellular (E) conformations. The I conformation has two populations with one interacting with the intracellular (H168) and/or extracellular pH sensors (?), directly or indirectly, depending on their protonation states. Decoupling of the S4 segment from pH sensors allows the spontaneous transitions toward the M and E conformations, with activating voltage mainly shifting the equilibrium of I–E transitions.

both labeling sites, under the same pH, only the transitions between the low and high FRET states (i.e., LH) exhibited voltage-dependent changes (*Figure 5A*). Activating voltage drove the equilibrium toward the high FRET state at the K125C-S224C sites and toward the low FRET state at the K169C-Q194C labeling sites (*Figure 5A*). At 0 mV and extracellular pH 7.5, again only the equilibrium of the LH transitions was remarkably shifted by intracellular pH, lowering it to 6.5 enriched the high FRET state at the K125C-S224C sites and the low FRET state at the K169C-Q194C labeling sites (*Figure 5A*). These data match well with functional results, that is lowering intracellular pH promotes hHv1 channel openings. We further showed that the H168Q mutation impaired the sensitivity of hHv1 channels to intracellular pH by diminishing the pH-dependent changes in $K_{eq}$ of LH transitions. Based on these kinetic data, we propose a kinetic model that explained voltage–pH gating in hHv1 channels (*Figure 5B*). Based on our smFRET data, the S4 segment may transit among three positions across the membrane, that is the intracellular side (I), middle (M), and extracellular side (E). The I conformation has two populations with their equilibrium depending on the protonation states of the intracellular pH sensor H168 and extracellular pH sensor that is yet to be identified (*Figure 5B*). Activating voltage mainly shifts the I–E transition by driving the S4 segment from the intracellular side (I conformation) directly to the extracellular side (E conformation), thereby promoting channel openings.

In summary, our results suggest that both pH and voltage act on the S4 segment of the hHv1 voltage sensor to modulate channel gating, which is consistent with the most recent gating current recordings (*Carmona, 2021*). In hHv1 channels, the mobilizing voltage of the S4 segment is probably much more negative, like those from canonical voltage-gated ion channels. However, the pH sensors at the intra- (H168) and the extracellular sides may physically interact with the S4 segment, directly or indirectly, to stabilize it at the inward resting conformations. The molecular couplings between the S4 segment and pH sensors are dependent upon the protonation state of the H168 and other pH-sensing residues. Thus, the stability of the inward resting conformation (i.e., I) is tuned by pH gradient, which in turn shifts the equilibrium of LH transition that is driven by voltages to open or close hHv1 channels.

## Discussion

Structural alignments of the voltage sensors from different organisms or even the same channels largely agree on a model where the positively charged S4 segment is driven outwards by depolarization – a movement utilized by pore modules to control ion permeation (*Guo et al., 2016*; *Li et al., 2014*; *Wisedchaisri et al., 2019*). However, the details of this movement are far from clear and, different movements have been described for different voltage sensors. For example, the S4 segment moves up for three helical turns in a voltage-gated sodium channel NavAb (*Wisedchaisri et al., 2019*) while it moves up for only one helical turn in a voltage-sensitive phosphatase (*Li et al., 2014*). Unfortunately, most atomic structures of voltage-gated ion channels were obtained at 0 mV voltage with their gating states unclear, some even at the conformations that may not be physiologically relevant (*Villalba-Galea et al., 2008*). Thus, the potential conformational changes of the voltage sensors inferred from the structural alignments remain to be functionally defined. We performed quantitative analysis on the real-time conformational transitions and dynamics of the hHv1 voltage sensor in lipid environments, at the voltages and pHs where the hHv1 channel functional status has been clearly defined. Our results showed that both voltage and pH act on the S4 segment in hHv1 channels to regulate channel gating. With smFRET imaging, we revealed the spontaneous, large conformational transitions of the S4 segment in hHv1 channels, which perhaps are the structural basis of the gating current noise (*Sigg et al., 1994*). The gating currents of Hv channels suggested multiple closed states (*Carmona et al., 2018*; *DeCoursey, 2018b*). Consistently, our data showed that the S4 segment in hHv1 channels transits among at least three conformations and it seems that only the transitions between the inward and outward conformations correlate with voltage and pH gating. Although hHv1 channels at deactivated and activated states are NOT available, long-time simulations have provided models of 'deactivated' and 'activated' conformations (*Geragotelis et al., 2020*). FRET values were predicted from these hHv1 structural models using the FRET Positioning and Screening (FPS) software and then compared with our data (*Kalinin et al., 2012*). From the 'deactivated' to 'activated' conformations, FPS software predicted the FRET changes from 0.28 to 0.57 at the K125C-S224C labeling sites, and from 0.72 to 0.45 at the K169C-Q194C labeling sites. The predicted FRET matched well with two of the three FRET centers identified by our smFRET studies, that is 0.27 and 0.6 at the K125C-S224C sites, and 0.78

and 0.52 at the K169C-Q194C sites. Interestingly, the unmatched FRET 0.9 state of K125C-S224C and FRET 0.24 state of K169C-Q194C sites, represented the same activated conformation (i.e., the E conformation in *Figure 5B*). Our results imply that the simulation study perhaps did not reach the fully activated conformations.

A previous study by Li et al. showed that the oligomeric state of purified hHv1 channels is concentration dependent with an apparent dissociation constant of 3.2 µM (*Li et al., 2015*). For smFRET samples, the hHv1 channels were reconstituted at an extremely low protein lipid ratio of 1:4000 (wt/wt) therefore the hHv1 concentration in liposome reconstitution was only ~38 nM, almost 100× lower than its dissociation constant. As a result, most hHv1 channels imaged were monomers and very few homodimers were excluded because they contained multiple donor and acceptor fluorophores, therefore exhibiting multiple bleaching steps. Electrophysiological results indicated that both monomeric and homodimeric hHv1 channels are functional, but the dimeric hHv1 channel exhibits strong intersubunit cooperativity by showing sigmoidal gating dynamics (*Smith and DeCoursey, 2013*). Therefore, it would be very interesting to further examine how subunit interactions alter the structural dynamics of hHv1 channels and, thus their gating kinetics. In the present work, we introduced the N214R background mutation to abolish the proton uptake into liposomes driven by positive voltages (*Figure 1— figure supplement 2*) that could change the extracellular pH environment of hHv1 channels. However, due to the technical challenge, we were not able to directly measure or even better, monitor the inside pHs and voltages of the liposomes that contain each individual hHv1 molecule being imaged. This limitation may impact the smFRET data collected at 0 mV with intra- and extraliposomal pH of 7.5 and 8.5 (*Figure 3A, B*, left panels, *Figure 4A, B*, middle panels). Under the above voltage/pH conditions, protons may flow out of the liposomes following the pH gradient through hHv1 channels that may still be activated, since the N214R mutation does not abolish the inward proton currents (*Carmona et al., 2018*; *De La Rosa and Ramsey, 2018*). The increase in extracellular pH of hHv1 channels due to proton efflux could attenuate, but probably does not cancel the effects of intracelullar pH 8.5, since enrichments of the FRET states representing resting conformations were still observed under the pH–voltage condition (*Figure 3A, B*, left panels, *Figure 4A, B*, middle panels).

In conclusion, the unique voltage and pH gating in the hHv1 channel is based on the spontaneous, voltage-dependent structural transitions of the S4 segment among multiple conformations. The intra- and extracellular pHs modify the stability of inward resting conformations by altering the protonation states of the pH-sensing residues that may directly or indirectly interact with the S4 segment in hHv1 channels. The gating model we proposed also explains the most recent patch-clamp fluorometry data and gating current from Hv channels (*Figure 5B*), which showed that the conformation transitions of the S4 segment are dependent on both intra- and extracellular pHs (*Carmona, 2021*; *Schladt and Berger, 2020*).

## Materials and methods
### Protein expression, purification and fluorophore labeling
The hHv1 cDNA was codon optimized and synthesized by Genscript Inc, then inserted into the pET28a(+) vector between the NdeI and BamHI restriction sites. The resulting hHv1 protein contained a 6× histidine tag at N-terminus, followed by a thrombin protease cutting site. All mutations were introduced into hHv1 by site-direct mutagenesis (Agilent Inc) and confirmed by DNA sequencing. The hHv1 proteins were expressed in *E. coli* BL21 (DE3) host cells and purified as that described by *Li et al., 2015*. In brief, the hHv1 proteins were purified by metal affinity chromatography and then loaded onto a Superdex-200 size-exclusion column with a running buffer containing 20 mM Tris, 150 mM NaCl, 1 mM Fos-Choline 12, and 1 mM TCEP (tris(2-carboxyethyl)phosphine), pH 8.0. For smFRET imaging samples, the proteins were subjected for buffer exchange using a 5 ml Hi-Trap desalting column to labeling buffer containing 20 mM HEPES (4-(2-hydroxyethyl)-1-piperazineethan esulfonic acid), 150 NaCl, 1 mM Fos-Choline 12 (*n*-dodecylphosphocholine), pH 7.0, and then mixed with Cy3 and Cy5 c5 maleimide (*Zheng et al., 2017*) in a 1:1 mixture at a protein fluorophore molar ratio of 1:6. The labeling reactions were performed under 4°C for 3 hr, then the free fluorophores were completely removed by performing a metal affinity chromatography and a size-exclusion chromatography in a row. All the proteins were either stored at −80°C for later use or reconstituted immediately.

## hHv1 liposome reconstitution

The lipids dissolved in chloroform containing POPE/POPG (3:1, wt/wt, Avanti Polar Lipids Inc) were dried in clean glass tubes with argon gas first, and then in a vacuum desiccator for 4 hr to evaporate the organic solvent completely. The dried lipids were resuspended in reconstitution buffer containing 20 mM HEPES, 5 mM KCl, 150 mM NaCl, pH 7.5 (for smFRET imaging) or 20 mM HEPES, 150 mM KCl, 0.05 mM NaCl, pH 7.5 (for liposome flux assay). The liposomes were formed by sonication and then destabilized with 10 mM Fos-choline 12 (Anatrace Inc). For liposome flux assay, the purified protein was mixed with the lipid solution at a protein lipid ratio of 1:200 (wt/wt), and the hHv1 proteolipo- somes were formed by detergent removal using Bio-beads SM2 (Bioad Inc). For smFRET imaging, the proteins were mixed with lipids at a ratio of 1:4000 (wt/wt) and the detergents were removed by dialyzing against reconstitution buffer containing 1 mM TCEP at the volume ratio of 1:500 for at least three times, each time over 12 hr.

## Liposome fluorescence flux assay

The $K^+$ gradient between inside and outside of the liposomes was established by diluting the liposomes in the extraliposomal buffer containing 20 mM HEPES, 150 mM NMDG (N-Methyl-D-glucamine), pH 7.5. The liposomes were incubated with 2 µM of ACMA fluorescence probes for ~5 min, then ACMA fluorescence was measured using a 96-well plate reader (FluoStar, Ex/Em = 390/460 nm) for ~5 min. After valinomycin was added at a final concentration of 0.45 µM, the fluorescence measurements were resumed with the same optical setting for ~40 min. Proton-specific ionophore CCCP was used as the positive control, and empty liposomes without hHv1 channels were used as negative controls. The fluorescence liposome flux data were processed following the method of *Su et al., 2016*. In brief, the hHv1 channel activities were calculated from the fluorescence readings normalized between 0 and 1 using the following equation:

$$A = (F_0 - F_{val})/(F_0 - F_{cccp})$$

where $F_0$, $F_{val}$, and $F_{cccp}$ were the steady-state ACMA fluorescence at initial, after adding valinomycin and CCCP, respectively. The relative activities of the hHv1 channel were normalized against the hHv1 WT proteoliposomes included in every batch of assays.

## Single-molecule imaging and data analysis

Flow chambers for smFRET imaging were prepared following the protocol of *Joo and Ha, 2012*. An objective-based TIRF built on a Nikon TE-2000U inverted microscope (TE-2000s) with ×100 APO TIRF NA1.49 objective lens, 532 and 640 nm lasers, was used for single-molecule imaging. Donor and acceptor emissions were separated by W-view Gemini beam splitter with chromatic aberration correction (Hamamatsu Inc) carrying the 638 nm long-pass beam splitter, then cleaned by 585/65 and 700/75 nm bandwidth filters (Chroma Inc). The images were collected by an ImagEM X2 EMCCD camera (Hamamatsu Inc). The liposomes containing fluorophore-labeled hHv1 channels were retained on the PEGlyated surface coated with biotinylated Histag antibodies (1:200 dilution, Thermo Fisher). Fluorophores were excited by a 532 nm laser (~1.0 W/cm²) and time-lapse movies were collected at 10 frames per second (i.e., time resolution of 100ms). The 640 nm laser (~1.0 W/cm²) was only used to confirm the existence of acceptor fluorophores when overall FRET was very low. Typical recording time was ~3 min, with half bleaching time being ~1 min. All imaging buffers contained ~3 mM Trolox, 5 mM PCA, and 15 µg/µl of PCD to enhance the photostability of the fluorophores (*Aitken et al., 2008*; *Dave et al., 2009*). For symmetrical pH conditions, β-escin at a final concentration of 50 µM was used to permeabilize liposomes (*Fan and Palade, 1998*). At least three batches of independent smFRET imaging were performed for each sample/condition. The movies were imported into the SPARTAN software directly without any corrections (*Juette et al., 2016*). The donor and acceptor channels were aligned using 2D maps generated from TetraSpeck fluorescent beads (T7279, Invit- rogen). The molecules were identified as point-spread functions using the threshold method with a window size of 7 pixels. The smFRET traces were extracted and then preselected using the Autotrace function of the SPARTAN software with criteria setting as FRET Lifetime >50 frames, donor/acceptor correlation coefficient between −1.1 and 0.5, signal-to-noise ratio >8, background noise level <70, Cy3 blinks <4, and overlap molecules removed (*Juette et al., 2016*). The resulting traces were further

picked manually following the criteria described in the previous studies (*Wang et al., 2016*; *Wang et al., 2018*). The FRET efficiency E was calculated with the following equation:

$$E = (I_A - I_D * 0.07)/(I_A + I_D)$$

where $I_A$ and $I_D$ are intensities of the donor and acceptor fluorophores. The crosstalk value is 0.07, determined from a protein sample containing the donor fluorophore only.

Overall, there were ~500 molecules per field of view with ~100 molecules containing both acceptor and donor fluorophores. The bin size of all histograms and contour maps was 0.03 and FRET contour plots were generated from the smFRET data of the first 5 s. For kinetic analysis, a kinetic model containing four FRET states was used, with one state assigned to bleaching or blinking events as FRET close to 0 (*Figure 1F*). The FRET peak centers were fixed for all smFRET traces from the same labeling sites, with the K125C-S224C sites as 0.27, 0.6, and 0.9 and the K169C-Q194C site as 0.24, 0.52, and 0.78. These peak centers were determined by Gaussian fit to the FRET histograms from all smFRET traces of the same labeling sites (*Figure 1—figure supplement 3B*). The smFRET traces were idealized using the MPL algorithm included in the SPARTAN software, which allows model constraints and directly optimizes rate constants obtained at different experimental conditions (*Qin et al., 2000*).

## Acknowledgements

This work was funded by NIH grant 1R15GM137215-01 (SW) and the startup fund of UMKC. We would like to thank Decker Gates for his help on protein preparation and data analysis; Dr. Lejla Zubcevic at the Kansas University Medical Center and Dr. Xiaolan Yao at the University of Missouri-Kansas City for their help in reviewing and revising the manuscript.

## Additional information

### Funding

| Funder | Grant reference number | Author |
| --- | --- | --- |
| National Institutes of Health | 1R15GM137215-01 | Shizhen Wang |
| University of Missouri-Kansas City | Startup fund | Shizhen Wang |

The funders had no role in study design, data collection, and interpretation, or the decision to submit the work for publication.

### Author contributions

Shuo Han, Data curation, Formal analysis, Investigation, Methodology, Validation, Visualization, Writing - review and editing; Sophia Peng, Joshua Vance, Kimberly Tran, Nhu Do, Nauy Bui, Zhenhua Gui, Investigation; Shizhen Wang, Conceptualization, Data curation, Formal analysis, Funding acquisition, Investigation, Methodology, Project administration, Resources, Software, Supervision, Validation, Visualization, Writing - original draft, Writing - review and editing

### Author ORCIDs

Shuo Han ⬤ http://orcid.org/0000-0003-0358-1851
Sophia Peng ⬤ http://orcid.org/0000-0002-7434-5445
Shizhen Wang ⬤ http://orcid.org/0000-0003-1065-4756

### Decision letter and Author response

Decision letter https://doi.org/10.7554/eLife.73093.sa1
Author response https://doi.org/10.7554/eLife.73093.sa2

## Additional files

### Supplementary files
• Transparent reporting form

### Data availability
The source data of all smFRET traces, contour maps, histograms, as well as liposome flux assay data is deposited in Dryad (Dryad Digital Repository, doi:10.5061/dryad.dv41ns1zs), including Figure 1b, c, d, e, g, h; Figure 2; Figure 3a and b; Figure 4a, b; Figure 5a, Figure 1—figure supplement 1a and c, Figure 1—figure supplement 2a, b, c; Figure 1—figure supplement 3a, b; Figure 1—figure supplement 4a and b.

The following dataset was generated:

| Author(s) | Year | Dataset title | Dataset URL | Database and Identifier |
|-----------|------|---------------|-------------|-------------------------|
| Wang S | 2022 | Data from: Structural dynamics determine voltage and pH gating in human voltage-gated proton channel | http://dx.doi.org/10.5061/dryad.dv41ns1zs | Dryad Digital Repository, 10.5061/dryad.dv41ns1zs |

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
