## [Editor Report]

This study examines the influence of voltage on conformational dynamics of voltage-sensing Hv1 channel at a single molecule resolution. Previously it was shown that although Hv1 channels lack a separate pore domain unlike most members of the voltage-gated channel family, the pore opening and voltage-sensing are distinct but linked processes. This study provides new insight in the mechanism of gating by showing that the voltage-sensor is able to access an intermediate conformation distinct from the activated and resting state.

---

## [Decision Letter]

**Decision letter after peer review:**

Thank you for sending your article entitled "Structural dynamics determine voltage and pH gating in human voltage-gated proton channel" for peer review at *eLife*. Your article is being evaluated by 3 peer reviewers, and the evaluation is being overseen by a Reviewing Editor and Richard Aldrich as the Senior Editor.

Please pay attention to Editor's note in parenthesis for further clarification.

*Reviewer #1:*

This paper uses a powerful single-molecule fluorescence approach to study the conformational dynamics of the voltage sensor in a human voltage-gated protein channel. The approach is impressive and builds on previous work of the senior author adapting single-molecule fluorescence to study gating dynamics. The major advance of the paper is the demonstration of smFRET on reconstituted human voltage gated proton (hHv1) channels in liposomes. The major conclusions of the paper are: (1.) the voltage sensing S4 helix is dynamic and altered by both pH and voltage; (2.) An intermediate state exists between the activated and resting states; (3.) H140 residue is important for sensing extracellular pH. These conclusions are supported by the data presented. The work could be further strengthened through a more thorough analysis of their existing data to make their proposed gating model more quantitative and predictive. Overall, this work will be of interest to colleagues in the ion channel and smFRET communities.

Major comments:

1. The authors claim that conformational changes between deactivated and activated states are probabilistic, not deterministic. This is not necessarily true without information on dynamics, as one can imagine a model where transitions from deactivated to activated proceed through an irreversible intermediate state (thereby, 3 states exist but transitions are deterministic). To test this hypothesis, the authors should examine the transitions between the states using transition density plots or (even better) build kinetic model(s) of the data as a function of voltage/pH using software such as QuB (which is accessible directly from SPARTAN) or ebFRET. Such analysis would allow rates to be included in Figure 5 and enable probabilities of state occupancies and transitions to be calculated given a pH or voltage, leading to testable predictions of the model in other backgrounds and for other voltage-gated channels.

2. A key advantage of smFRET data is the ability to link conformational dynamics to structure. While the authors estimate the movement of the S4 segment to be ~20 angstroms, it is unclear how large the movement to the "intermediate" state is, whether that distance is consistent between the different FRET pairs used, and whether that distance agrees with structural predictions.

3. The authors note the use of SPARTAN for single-molecule analysis; however, more details need to be provided to communicate how the data were treated to ensure reproducibility. For example, it is unclear how images were preprocessed, how single molecules were defined (e.g., pixel size), how crosstalk between the channels was accounted for (if at all) and if any background subtraction was performed. In addition, there is no equation provided for how FRET efficiency was computed and whether correction factors were applied (see DOI: 10.7554/*eLife*.60416 for example).

4. Two comments regarding single molecule idealization. First, the authors use SPARTAN and fit the data to three states, but it is unclear if any other models were tested (e.g., 2 states or 4 states) and if so, how they were ranked. This is particularly important since a major conclusion of the paper rests on the existence of a third, intermediate state. Second, it is unclear why transitions are identified in the Viterbi paths (idealized trajectories) once the acceptor has photobleached (FRET = 0) in Figures 1d and S3b. This suggests the model may be overfitting the data and interpreting noise as signal. Were the traces pre-processed prior to idealization by truncating at the photobleaching event?

*Reviewer #2:*

The manuscript entitled "Structural dynamics determine voltage and pH gating in human voltage-gated proton channel" by Shuo Han et al. studied the conformational dynamics of the human voltage-gated proton channel (Hv1) under different voltages and pHs, which are two stimuli that change the open probability of the channel. The authors purified the wild-type Hv1 and confirmed its function after reconstitution using fluorescence flux assays. Then, they introduced two pairs of cysteine mutations in the cysteineless background and measured the conformational dynamics through single-molecule fluorescence resonance energy transfer (smFRET) after labeling the channel. However, they did not observe changes in smFRET when voltage was changed in the proteoliposomes. Then, they introduced the N214R mutation, which abolised Hv1 proton currents without perturbing the channel's gating. In this background, smFRET population distributions were observed at different voltages, suggesting that the absence of changes found in the wild-type Hv1 was caused by the internal acidification of the liposomes, which disfavor the channel activation. Next, using the N214R background mutant, the authors showed that smFRET population distributions depended on the imposed transmembrane voltage. The observed smFRET changes at different voltages suggested that an S4 outward movement occurs at depolarizing voltages. The smFRET population distributions also depended on the internal and external pH values implying that the S4 position changed according to the pH values. The smFRET changes observed at different internal pHs were abolished in the H168Q mutant, which was previously shown to alter the internal pH dependence of the channel. Finally, they propose that H140 is the external pH sensor since the H140A mutation showed smFRET population distribution changes in the wild-type background channel. Although the approach is novel and the results are interesting, the work has a series of important issues that the authors must address.

– A main concern about this paper is that the authors must determine if the dimeric nature of Hv1 is affecting their experimental results. Since the Hv1 dimer is small, there is the possibility that two pairs of fluorophores in different subunits are close enough in space to produce FRET. When two pairs of cysteine mutations are inserted in the channel to measure smFRET between them, there are four cysteines per molecule instead of two. Therefore, the multiple levels of FRET signals measured by the authors could be produced by a putative complicated combination of distances between fluorophores located in different subunits. The authors must address this critical point to assure that the interpretation of the smFRET signal changes is correct.

– It seems that the authors have confusion regarding the activation of the Hv1 voltage sensor, which also applies to other voltage-gated ion channels and proteins in general. As any kinetic process, there are expected different protein conformations at the steady-state distribution, and the macromolecular observable is the average of the ensemble. The state's distribution and the macromolecular ensemble change when the system is perturbed, as it happens for Hv1 when the voltage or the pH is changed. During the entire manuscript, the authors claim that their results "… suggested the biological gating is determined by the conformational distributions of the hHv1 voltage sensor, rather than the conformational transitions between the presumptive 'resting' and 'activated' conformations." There are similar statements in all the sections of the manuscript. Structural models obtained from X-ray diffraction or cryo-EM are models of the more stable conformations, which does not mean that the atomic positions of the protein undergoing activation or deactivation gating are constrained to only those possible conformation trapped by X-ray crystallography. I think the authors' statements refer to the high structural dynamics of the channel, i.e., the small kinetics barriers between the states of the channel. The authors should modify these ideas or interpretations in the manuscript.

(Editor's Note- This comment has come up later too. Please rephrase these sentences to emphasize the novelty of the study. In our opinion, it is the detection of these low probability conformations for voltage-activated process. It may be worth calculating how this compares with a distribution predicted for a simple two state process, where the channel is either activated or resting)

– There is no quantifiable and statistical criterium in the manuscript to demonstrate differences or similarities of the measured distributions. Changes in the smFRET population distributions between different conditions seem subjective, and in some cases, the differences are not evident or clearly derived from the presented results. For instance, although the smFRET population distributions are evident in the mutant K169C-Q194C-N214R at different voltages in figure 1c, this is not obvious for the mutant K125C-S224C-N214R in figure 1e since the red density histogram in the figure does not correspond with the contour map showed. Similarly, figure 2 is poorly analyzed, and no statistical analysis is evident to demonstrate the validity of the conclusions. From the contour maps shown for the mutant S98C-Q194C-N214R, this reviewer observed a higher occupancy of low smFRET at -85 mV compared with 0 and 120 mV, especially when compared with the contours shown in figure 1e. Figure 3a has the same problem since distributions are different for the low smFRET when pHin/pHout 7.5/7.5 and 8.5/7.5 are compared, although the authors do not discuss that. As presented now, it seems that the authors observe differences only when it is convenient for the paper's narrative. A proper statistical analysis of the data is needed in the presented figures and the used methodology.

– Symmetrical pH distributions were obtained using β-escine (concentration? Time of incubation?), while the asymmetric pHs were obtained by changing the extra liposome solution pHs. A better approach would be to establish symmetrical pH during liposomes formation and avoid permeabilization with β-escine since it can perturb the Hv1 conformational dynamics.

(Editor's note- Please address in your discussion)

– Authors concluded that "… Our data conclusively indicate that pH gating of the hHv1 channels originates from the pH-dependent conformational changes of the voltage sensing S4 segment." The data shown by the authors is not sufficient to support this claim since only the mutant K125C-S224C-N214R was evaluated. To properly support this statement, the authors must measure the smFRET distributions using other mutants, including the mutant K169C-Q194C-N214R along with those shown in figure 2. This approach has the additional advantage of showing a better picture of the conformational changes of Hv1 at different pHs.

– To support the claim that H168 is the key internal pH sensor in hHv1, additional experiments are needed. I strongly suggest that electrophysiology should be performed to confirm this claim. Also, the authors' results do not support a direct interaction of H168 or H140 and S4, so it should be stated explicitly that this is only a proposed model to be tested in the future unless additional evidence is presented.

(This is controversial and may require additional experiments. See if you want to possibly resubmit a report rather than a full length article and remove the pH sensor part. We can discuss this if needed.)

– Proton fluxes assays presented in this work are very long when compared with results previously reported by others. For instance, the Mackinnon's group obtained fluorescence steady-state readings after 5 minutes of 40 nM valinomycin addition (Lee et al., 2009). In contrast, the steady-state level in this work is reached at around 30 minutes after adding 450 nM valinomycin, which is an exceedingly high concentration. This difference suggests to this reviewer that perhaps the functional integrity of the purified hHv1 channel has been compromised during purification process and/or liposome reconstitution. The authors should discuss the origin of this discrepancy with previously published work.

– Since proton flux measurements take so long despite of using very high concentrations of valinomycin, a control with empty liposomes treated with the same reconstitution protocol without hHv1 should be included to demonstrate that it is the channel that produced the observed proton fluxes and that they are not originated from liposomes' proton leak. The controls included in figure S1 are insufficient since leaky liposomes will show the same results even in the absence of protein.

– A proton flux assay of the mutant H140A in the wild-type hHv1 background must be included to assure the single mutation does not alter the hHv1 function.

– There is an inconsistency between fluxes shown in figure 4c and S2c. The former showed higher activity of mutant K169C-Q194C compared to wild-type, while the latest show wild-type has higher or equal activity than the mutant. A similar incongruency is evident between figure S1c and S2b in the wild-type channel fluxes. Representative flux assays of each mutant must be included in a supplementary figures section to show the constancy and reproducibility of the functional assay.

– Protein reconstitution protocols for the functional assay and smFRET measurements are different. Is the hHv1 function comparable between these protocols? The authors must demonstrate that the function of the proteoliposomes is similar when the protein is reconstituted using these two different protocols.

*Reviewer #3:*

The gating of Hv1 protons channels is distinguished from related voltage-sensitive ion channels and phosphatases by an apparently unique sensitivity to changes in both membrane potential and the transmembrane pH gradient (ΔpH = pHo – pHi). The mechanism of ΔpH sensitivity in Hv1 is of widespread interest, but remains only partially understood. In particular, extracellular ionizable residues that are postulated to be required for sensing changes in pHo remain unidentified.

Here the authors utilize a FRET approach in which recombinant, purified Hv1 channels carrying Cys mutations at pairs of extra- and intra-cellular residues are labeled with separate fluorophores and fluorescence changes (and the corresponding FRET ratios) are measured optically over time under conditions that are predicted to alter membrane potential and at various pHi. Although the development of a new optical assay to indirectly measure Hv1 channel activity represents a potentially important advance in the field, the data on which the authors base their main conclusions can be explained by alternative mechanisms that have not been ruled out. Furthermore, the authors' putative identification of a single mutation (H140A) that is purported to abolish pHo sensitivity is inconsistent with previous study showing that simultaneous mutation of both candidate extracellularly-exposed His residues in Hv1 (H140A-H193A) is insufficient to abolish pHo-dependent shifts in the apparent POPEN-V relation measured using voltage clamp electrophysiology 1.

(The previous electrophysiology data is particularly problematic. Please clarify how you expect this to be reconciled.)

The caveats to interpretation of the data presented in this manuscript are many, but several central concerns are outlined here:

1. FRET changes measured here are not independently correlated with channel activity measured using electrophysiological methods, and it remains unclear what effects the double-Cys mutations (either before or after labeling) may have had on the voltage dependence and/or kinetics of channel gating. The reader must assume that mutant channels are WT-like, but a previously published survey of many Hv1 mutations clearly shows that altering channel structure can have major impacts on functional parameters 1.

(Address the caveats in discussion..)

2. The authors interpret FRET changes to selectively report movement of the S4 helix, but Cys residues are introduced into predicted loops between helical segments, and could also report local changes that occur independently of S4 helix movements that are thought to underlie voltage-dependent gating. There appears to be no independent experimental evidence that FRET changes actually report S4 movement. Of concern, only one mutant pair (K169C-Q194C) appears to exhibit a substantial voltage-dependent change in FRET ratio (5 other Cys mutant pairs exhibit little or no apparent voltage-dependent FRET changes), and data in Figure 1f showing this change contains undescribed error bars; neither is any statistical method for quantifying the magnitude of the voltage-dependent FRET change described in Methods.

(This type of concern is understandable but the fact that you are measuring conformational distributions indicates that the channel S4 is detecting different conformations although it may be hard to predict the specific details of this conformation)

3. The data shown in Figure 4d are interpreted to mean that H140A mutation abolishes pHo sensitivity measured by pH-dependent quenching of a pH-sensitive dye (ACMA) within liposomes containing purified and reconstituted Hv1 channels (a similar method was previously described 2, but is evidently not cited in the References). The authors state that "on the H140A mutation background, the FRET distributions at the K169C-Q194C labeling sites do shift by voltage (Figure 4d)", which seems to mainly argue that this mutant channel remains voltage-dependent, and evidently does not directly address whether pHo sensitivity is altered. Furthermore, a previous study showed that H140A-H193A mutant Hv1 channels retain WT-like pHo sensitivity (-46 mV/pHo unit shift in VTHR, which is similar to the -38 mV/pHo unit shift measured for WT Hv1; see Table S1) 1.

1 Ramsey, I. S. et al. An aqueous H^+^ permeation pathway in the voltage-gated proton channel Hv1. Nat Struct Mol Biol 17, 869-875, doi:10.1038/nsmb.1826 (2010).

2 Lee, S. Y., Letts, J. A. & MacKinnon, R. Functional reconstitution of purified human Hv1 H^+^ channels. J Mol Biol 387, 1055-1060, doi:10.1016/j.jmb.2009.02.034 (2009).

Recommendations for the authors:

1. Determine whether mutations alter biophysical properties of expressed Hv1 channels.

2. Establish that FRET changes necessarily report conformational rearrangements of the S4 helix, and not other motions that may be correlated with but not causal to voltage-dependent gating.

3. Address why FRET data are evidently contradictory to previously published electrophysiological data.

[Editors' note: further revisions were suggested prior to acceptance, as described below.]

Thank you for resubmitting your work entitled "Structural dynamics determine voltage and pH gating in human voltage-gated proton channel" for further consideration by *eLife*. Your revised article has been evaluated by Richard Aldrich (Senior Editor) and a Reviewing Editor.

The manuscript has been improved but there are some remaining issues that need to be addressed in the text, as outlined below:

1. Overall, the authors basic observation that changes in membrane potential alter S4 position in a way that is consistent with the general consensus of voltage sensor function and previous reports of Hv1 specifically. Despite the potential of smFRET to reveal new insights into conformational rearrangements that occur during Hv1 channel gating, caveats to the interpretations of the data limit the value of the information in the current manuscript, and the work does not significantly advance knowledge in the field.

Data in Figure 3 lead the authors to hypothesize that in H168Q, more negative voltage is required to move S4 into the resting (down) conformation than would be required for WT Hv1. A previous report1 shows that whereas both WT Hv1 and H168T manifest qualitatively similar positive shifts in the positions of their G-V curves as pHi is raised, the magnitude of the pH sensitive-shift in the position of the G-V curve is shallower (~20 mV/pHi unit) in H168T than in WT Hv1 (~40 mv/pHi unit). A positive shift in the G-V (i.e., at higher pHi) shows that channel closing occurs at less hyperpolarized potentials (or opening occurs at more depolarized voltages), which is opposite to the conclusion reached by the authors. One possible explanation is that the direction of proton currents at positive potentials is inward for H168T at high pHi all voltages between Vthr and the Nernst potential for H^+^ (EH^+^). Because the G-V is less shifted at acidic pHi in H168 mutants, there is a higher likelihood that the net current is inward, and if proton fluxes are oppositely directed in WT and H168Q channels, one might expect dramatically different effects on intra-liposomal pH (pHo) to result, as demonstrated by the authors for WT vs. N214R.

The apparent effects of pHi acidification on S4 position is complicated by the possibility pHi is unknown under the experimental conditions used here.

Please discuss these concerns in the discussion.

2. An alternative explanation for the data in Figure 3 not stated by the authors is that Hv1 H168Q channels enter a closed-state conformation in which nonetheless S4 remains in the activated (out) position, but this seems unlikely given that Hv1 channels do not appear to inactivate. The apparent paradox requires explanation and/or demonstration that membrane potentials and intra-liposomal pH remain intact during the experiments with H168Q; similar caveats apply to measurements of fluorescently-labeled WT (K125C-S224C and K169C-Q194C) channels. Unfortunately, it's not clear to me that the authors will be able to measure membrane potential and/or intra-liposomal pH (pHo) may not be within the authors av

Cherny, V. V., Morgan, D., Thomas, S., Smith, S. M. E. & DeCoursey, T. E. Histidine(168) is crucial for DeltapH-dependent gating of the human voltage-gated proton channel, hHV1. J Gen Physiol, doi:10.1085/jgp.201711968 (2018).

In order to provide a nuanced perspective, please discuss the above limitations and its impact on your conclusions.

3. There are still a number of typos/mistakes in the revised version. Please have it proofread carefully if needed by an experienced colleague.

---

## [Author Response]

Reviewer #1:This paper uses a powerful single-molecule fluorescence approach to study the conformational dynamics of the voltage sensor in a human voltage-gated protein channel. The approach is impressive and builds on previous work of the senior author adapting single-molecule fluorescence to study gating dynamics. The major advance of the paper is the demonstration of smFRET on reconstituted human voltage gated proton (hHv1) channels in liposomes. The major conclusions of the paper are: (1.) the voltage sensing S4 helix is dynamic and altered by both pH and voltage; (2.) An intermediate state exists between the activated and resting states; (3.) H140 residue is important for sensing extracellular pH. These conclusions are supported by the data presented. The work could be further strengthened through a more thorough analysis of their existing data to make their proposed gating model more quantitative and predictive. Overall, this work will be of interest to colleagues in the ion channel and smFRET communities.Major comments:1. The authors claim that conformational changes between deactivated and activated states are probabilistic, not deterministic. This is not necessarily true without information on dynamics, as one can imagine a model where transitions from deactivated to activated proceed through an irreversible intermediate state (thereby, 3 states exist but transitions are deterministic). To test this hypothesis, the authors should examine the transitions between the states using transition density plots or (even better) build kinetic model(s) of the data as a function of voltage/pH using software such as QuB (which is accessible directly from SPARTAN) or ebFRET. Such analysis would allow rates to be included in Figure 5 and enable probabilities of state occupancies and transitions to be calculated given a pH or voltage, leading to testable predictions of the model in other backgrounds and for other voltage-gated channels.

We have performed kinetic analyses on the smFRET data at different labeling sites under different voltage/pH.

a) First we combined all the smFRET data from the same labeling sites and performed multiple Gaussian fits (1~5) to the FRET histograms. The SSE (sum of squared errors) of curve fittings decreases by >95% with 3 FRET states for histograms of both K125C-S224C and K169C-Q194C labeling sites (Figure 1, figure supplement 3B). Therefore, we decided to analyze the smFRET data using a kinetic model containing 3+1 FRET states. The additional one state with FRET close to 0 was included for occasional blinking or bleaching events included by the SPARTAN software by mistake (Line 125-134).

b) We combined all smFRET data of the same labeling sites and then determined the centers of 3 FRET states by performing Gaussian curve fittings with their histograms, which are 0.27, 0.6 and 0.9 for the K125C-S224C labeling sites, 0.24, 0.52 and 0.78 for the K169C-Q194C labeling sites. For all kinetic analyses, the centers of FRET states were manually fixed so the changes in the FRET distributions induced by voltage and/or pH can be compared fairly.

c) With the 4 FRET state model, we performed kinetic analyses using the Maximum Point Likelihood (MPL) algorithm. The MPL algorithm optimizes the rate constants directly and allows model constraints (i.e fixing FRET centers in our analyses). With the MPL algorithm, we can analyze multiple smFRET data sets at different pH/voltage and compare the distributions and rates fairly.

d) We idealized the smFRET traces based on the 4 FRET state model (Figure 1F) using the MPL algorithm and included the state occupancies at different voltage/pH in the revised manuscript. We further calculated the equilibrium constants (i.e the ratios of the forward K_F_ and reverse K_R_ transition rates) at different voltages. The results suggest that voltage/pH drives primarily the transitions between the low and high FRET states at both the K125CS224C and K169C-Q194C labeling sites (Line 212-233, Figure 5A and B). Together with transition density plots (Line 143-146, Figure 1G and H), our data suggested that activating voltage and low intracellular pH mainly drive the S4 segment from the intra- to extracellular side to promote channel opening.

2. A key advantage of smFRET data is the ability to link conformational dynamics to structure. While the authors estimate the movement of the S4 segment to be ~20 angstroms, it is unclear how large the movement to the "intermediate" state is, whether that distance is consistent between the different FRET pairs used, and whether that distance agrees with structural predictions.

So far, no high resolution structures of Hv channels at resting and activated states are available for comparison. But a recent long time molecular dynamics simulation study has provided the hHv1 structural models at potential ‘resting’ and ‘activated’ states (Geragotelis A et al., PNAS 2020). Using the FRET Positioning and Screening (FPS) software developed by Kalinin et al. (Nature Methods 2012), we calculated the accessible volumes of the donor and acceptor fluorophores at the K125C-S224C, K169C-Q194C labeling sites and predicted their FRET values at the ‘resting’ and ‘activated’ states. The following table included the FRET peaks identified by our smFRET data and the mean FRET values predicted by the FPS software.

As shown in Author response table 1, transiting from ‘resting’ to ‘activated’ conformations, the FPS software predicted FRET changes from 0.28 to 0.57 at the K125C-S224C labeling sites, from 0.72 to 0.45 at the K169C-Q194C labeling sites. The predicted FRET match well with 2 of 3 FRET peaks identified by our smFRET data, at both labeling sites. Interestingly, the unmatched 0.9 FRET peak of the K125C-S224C site and 0.24 FRET peak of the K169C-Q194C sites represent the same conformation of the S4 segment at the extracellular side that is enriched by activating voltages. The data suggest that the S4 segment in the ‘activated’ structure from the simulation study perhaps does not reach the fully activated state yet.

**Author response table 1. sa2table1:** 

	K125C-S224C	169C-Q194C				
smFRET	0.27	0.6	**0.9**	**0.24**	0.52	0.78
FPS	0.28 (R)	0.57(A)			0.45 (A)	0.72 (R)

We have included the above analyses and discussions in the revised manuscript (Line 264275).

3. The authors note the use of SPARTAN for single-molecule analysis; however, more details need to be provided to communicate how the data were treated to ensure reproducibility. For example, it is unclear how images were preprocessed, how single molecules were defined (e.g., pixel size), how crosstalk between the channels was accounted for (if at all) and if any background subtraction was performed. In addition, there is no equation provided for how FRET efficiency was computed and whether correction factors were applied (see DOI: 10.7554/eLife.60416 for example).

We have included all the details of the single-molecule analysis in the revised manuscript (Line 364-367 and 371-394).

In brief, the images were directly imported into SPARTAN software without preprocessing and smFRET traces were extracted following the standard settings. The donor and acceptor channels were aligned using 2D maps generated from the tetraSpec 0.2 μm bead samples. The window size of 7 pixels was used to identify individual molecules and calculate their donor and acceptor intensities. The crosstalk value of 0.07 was applied which was determined using a donor fluorophore only sample. FRET efficiencies were calculated using the following equation without any other corrections.

FRET = (I_A_-0.07*I_D_)/(I_D_+I_A_)

Where I_D_ and I_A_ are the intensities of donor and acceptor fluorophores, respectively.

4. Two comments regarding single molecule idealization. First, the authors use SPARTAN and fit the data to three states, but it is unclear if any other models were tested (e.g., 2 states or 4 states) and if so, how they were ranked. This is particularly important since a major conclusion of the paper rests on the existence of a third, intermediate state. Second, it is unclear why transitions are identified in the Viterbi paths (idealized trajectories) once the acceptor has photobleached (FRET = 0) in Figures 1d and S3b. This suggests the model may be overfitting the data and interpreting noise as signal. Were the traces pre-processed prior to idealization by truncating at the photobleaching event?

As we have responded in 1(a), we combined all the FRET data from the same labeling sites and then performed Gaussian fits with 1-5 states. We found the SSE (Figure 1—figure supplement 3B) of fits are decreased by >95% with 3 FRET states, thus determined to use a 3 state model to analyze all the FRET data of the K125C-S224C and K169C-Q194C labeling sites.

The SPARTAN software contains algorithms that can identify most of the bleaching and blinking events correctly and automatically remove the data points after fluorophore bleaches. However, it occasionally fails to exclude a few data points after fluorophore was bleached. Therefore, we introduced the F_B_ state in the kinetic model with FRET as 0 to eliminate their impacts on kinetic analysis. As shown in Figure 1G, H, the blinking or bleaching events included are rare and almost invisible in transition density plots. Therefore, the occupancies and rates of the F_B_ state were ignored in the following kinetic analyses.

Reviewer #2:The manuscript entitled "Structural dynamics determine voltage and pH gating in human voltage-gated proton channel" by Shuo Han et al. studied the conformational dynamics of the human voltage-gated proton channel (Hv1) under different voltages and pHs, which are two stimuli that change the open probability of the channel. The authors purified the wild-type Hv1 and confirmed its function after reconstitution using fluorescence flux assays. Then, they introduced two pairs of cysteine mutations in the cysteineless background and measured the conformational dynamics through single-molecule fluorescence resonance energy transfer (smFRET) after labeling the channel. However, they did not observe changes in smFRET when voltage was changed in the proteoliposomes. Then, they introduced the N214R mutation, which abolised Hv1 proton currents without perturbing the channel's gating. In this background, smFRET population distributions were observed at different voltages, suggesting that the absence of changes found in the wild-type Hv1 was caused by the internal acidification of the liposomes, which disfavor the channel activation. Next, using the N214R background mutant, the authors showed that smFRET population distributions depended on the imposed transmembrane voltage. The observed smFRET changes at different voltages suggested that an S4 outward movement occurs at depolarizing voltages. The smFRET population distributions also depended on the internal and external pH values implying that the S4 position changed according to the pH values. The smFRET changes observed at different internal pHs were abolished in the H168Q mutant, which was previously shown to alter the internal pH dependence of the channel. Finally, they propose that H140 is the external pH sensor since the H140A mutation showed smFRET population distribution changes in the wild-type background channel. Although the approach is novel and the results are interesting, the work has a series of important issues that the authors must address.*–* A main concern about this paper is that the authors must determine if the dimeric nature of Hv1 is affecting their experimental results. Since the Hv1 dimer is small, there is the possibility that two pairs of fluorophores in different subunits are close enough in space to produce FRET. When two pairs of cysteine mutations are inserted in the channel to measure smFRET between them, there are four cysteines per molecule instead of two. Therefore, the multiple levels of FRET signals measured by the authors could be produced by a putative complicated combination of distances between fluorophores located in different subunits. The authors must address this critical point to assure that the interpretation of the smFRET signal changes is correct.

Our smFRET studies were performed on monomeric, not homodimeric hHv1 channels. As shown by Li and Perozo et al. (PNAS 2015), the oligomeric states of purified hHv1 channels are concentration-dependent, with an apparent disassociation constant of ~3.2 uM. To prepare liposomes for smFRET studies, the fluorophore-labeled hHv1 proteins were mixed with lipids at a protein lipid ratio of 1:4,000 (w/w), and the lipid concentration is only 5 mg/ml. Therefore the hHv1 protein concentration during liposome reconstitution is only ~38 nM, 100 folds lower than the Kd, therefore most hHv1 channels will be monomeric.

Given that the labeling efficiencies of all hHv1 mutants were close to 100%, we found that most hHv1 molecules imaged only contain two fluorophores (either 2 donors, 2 acceptors, or 1 donor/acceptor pair), also suggesting that hHv1 channels indeed exist mainly as monomers. Moreover, some molecules containing >2 fluorophores, from either homodimeric or aggregated hHv1 channels, were rejected during the automatic and manual trace picking processes for they exhibit more than 1 donor or acceptor bleaching step.

As a result, the smFRET data we collected and analyzed were essentially from monomeric hHv1 channels at the designed labeling sites. We have clarified this in the manuscript (Line 276-286).

However, the reviewer indeed raised a great point about the potential differences in structural dynamics of monomeric and homodimeric hHv1 channels. Electrophysiological results indicated that both monomeric and homodimeric hHv1 channels are functional, but the dimeric channels exhibit strong intersubunit cooperativity and therefore sigmoidal gating dynamics. Although it is not within the scope of the current manuscript, we have started to examine the structural dynamics of dimeric hHv1 channels to understand how intersubunit interactions alter structural dynamics and thus gating dynamics.

– It seems that the authors have confusion regarding the activation of the Hv1 voltage sensor, which also applies to other voltage-gated ion channels and proteins in general. As any kinetic process, there are expected different protein conformations at the steady-state distribution, and the macromolecular observable is the average of the ensemble. The state's distribution and the macromolecular ensemble change when the system is perturbed, as it happens for Hv1 when the voltage or the pH is changed. During the entire manuscript, the authors claim that their results "… suggested the biological gating is determined by the conformational distributions of the hHv1 voltage sensor, rather than the conformational transitions between the presumptive 'resting' and 'activated' conformations." There are similar statements in all the sections of the manuscript. Structural models obtained from X-ray diffraction or cryo-EM are models of the more stable conformations, which does not mean that the atomic positions of the protein undergoing activation or deactivation gating are constrained to only those possible conformation trapped by X-ray crystallography. I think the authors' statements refer to the high structural dynamics of the channel, i.e., the small kinetics barriers between the states of the channel. The authors should modify these ideas or interpretations in the manuscript.(Editor's Note- This comment has come up later too. Please rephrase these sentences to emphasize the novelty of the study. In our opinion, it is the detection of these low probability conformations for voltage-activated process. It may be worth calculating how this compares with a distribution predicted for a simple two state process, where the channel is either activated or resting)

We appreciate the reviewer and editor’s suggestions, these interpretations were removed in the revised manuscript. The kinetic analyses in the revised manuscript have provided new insights into how voltage and pH shift the conformational transitions of the S4 segment of hHv1 channels (Line 212-233).

– There is no quantifiable and statistical criterium in the manuscript to demonstrate differences or similarities of the measured distributions. Changes in the smFRET population distributions between different conditions seem subjective, and in some cases, the differences are not evident or clearly derived from the presented results. For instance, although the smFRET population distributions are evident in the mutant K169C-Q194C-N214R at different voltages in figure 1c, this is not obvious for the mutant K125C-S224C-N214R in figure 1e since the red density histogram in the figure does not correspond with the contour map showed. Similarly, figure 2 is poorly analyzed, and no statistical analysis is evident to demonstrate the validity of the conclusions. From the contour maps shown for the mutant S98C-Q194C-N214R, this reviewer observed a higher occupancy of low smFRET at -85 mV compared with 0 and 120 mV, especially when compared with the contours shown in figure 1e. Figure 3a has the same problem since distributions are different for the low smFRET when pHin/pHout 7.5/7.5 and 8.5/7.5 are compared, although the authors do not discuss that. As presented now, it seems that the authors observe differences only when it is convenient for the paper's narrative. A proper statistical analysis of the data is needed in the presented figures and the used methodology.

We appreciate the reviewer and editor’s suggestions. We have conducted statistical tests (unpaired two-tailed t-test) on the FRET state occupancies and included these results in figures so the changes of FRET distributions can be compared more objectively. The histograms were generated from ALL data points while contour maps only contained the data points from the FIRST 5 SECONDS. For clarity, we have moved the contour maps to the supplementary figures, for the main purpose of the contour maps is to show that the smFRET distributions reach equilibrium states and do not change over time.

– Symmetrical pH distributions were obtained using β-escine (concentration? Time of incubation?), while the asymmetric pHs were obtained by changing the extra liposome solution pHs. A better approach would be to establish symmetrical pH during liposomes formation and avoid permeabilization with β-escine since it can perturb the Hv1 conformational dynamics.

We added 50 μm β-escin in the imaging buffer and waited >5 min before collecting image data. Permeabilization of liposomes by β-escin is to ensure that not only pH but also other ions between the inside and outside of the liposomes are identical. The proton influx or efflux could happen with a very small ionic gradient (even buffer reagents) across liposomes, which may significantly alter the intraliposomal pH. In my previous work, β-escin was used (Wang et al., Nature Chemical Biology 2019) to study a potassium channel, and no visible structural effect by β-escin was observed.

(Editor's note- Please address in your discussion)– Authors concluded that "… Our data conclusively indicate that pH gating of the hHv1 channels originates from the pH-dependent conformational changes of the voltage sensing S4 segment." The data shown by the authors is not sufficient to support this claim since only the mutant K125C-S224C-N214R was evaluated. To properly support this statement, the authors must measure the smFRET distributions using other mutants, including the mutant K169C-Q194C-N214R along with those shown in figure 2. This approach has the additional advantage of showing a better picture of the conformational changes of Hv1 at different pHs.

We examined the pH effects with the K169C-Q194C-N214R mutant with and without the H168Q mutation and included the new data in the revised manuscript. The new data match well with these from the K125C-S224C labeling sites (Figure 3B, Figure 4B).

– To support the claim that H168 is the key internal pH sensor in hHv1, additional experiments are needed. I strongly suggest that electrophysiology should be performed to confirm this claim. Also, the authors' results do not support a direct interaction of H168 or H140 and S4, so it should be stated explicitly that this is only a proposed model to be tested in the future unless additional evidence is presented.

H168 as a major intracellular pH sensor has been well established by Cherny and DeCoursey et al. using electrophysiological studies (Cherny et al., JGP 2018). At this moment, we were not able to identify the interacting residues of H168 at the S4 segment, for lacking electrophysiological equipment. We have made revisions in the manuscript to clarify it (Line 188-190).

We also removed the results of H140 for the reasons explained above.

(This is controversial and may require additional experiments. See if you want to possibly resubmit a report rather than a full length article and remove the pH sensor part. We can discuss this if needed.)– Proton fluxes assays presented in this work are very long when compared with results previously reported by others. For instance, the Mackinnon's group obtained fluorescence steady-state readings after 5 minutes of 40 nM valinomycin addition (Lee et al., 2009). In contrast, the steady-state level in this work is reached at around 30 minutes after adding 450 nM valinomycin, which is an exceedingly high concentration. This difference suggests to this reviewer that perhaps the functional integrity of the purified hHv1 channel has been compromised during purification process and/or liposome reconstitution. The authors should discuss the origin of this discrepancy with previously published work.

According to Lee et al. (JMB 2010), the fraction of the hHv1 channels that are functional is reflected by F_Hv_/F_total_, where F_Hv_ and F_total_ are the fraction fluorescence quenching after adding valinomycin and CCCP, respectively. If the fraction of the functional hHv1 channels is close to 1, the ratio is ~0.8 at the protein/lipid ratio of 1/100 (Figure 1c in Lee et al). As shown in Figure 1—figure supplement 1C and F2B of our manuscript, the ratio is also ~0.8. So, nearly all the hHv1 channel proteins we studied are functional.

Another aspect of the hHv1 channel function is their relative channel activities, partly reflected by the slope (i.e speed) of ACMA fluorescence quenching after adding valinomycin. In the study of Lee et al., the ACMA fluorescence quenching typically reached a plateau within 1~ 2.5 min when the protein/lipid ratio was >1/5,000. In our liposome assays, the time for ACMA fluorescence quenching to reach a plateau is slightly different from batch to batch, ranging from 2 min (Figure s2b, bottom right panel) to 6 min (Figure s1c). In the assays of Lee et al. and ours, ACMA fluorescence quenching all exhibits slow liner drop after reaching the plateau. So, it seems that the relative activities of our hHv1 proteins are ~3 fold less than those prepared by Lee et al. We included 0.5 mg/ml BSA, 0.2 mM EGTA and 2 mM β-mercaptoethanol in the liposome flux assays but did not identify any differences. So other differences that may impact the relative activities of hHv1 channels include: (a) the expression system (eukaryotic *Pichia pastoris* vs bacterial *E. coli*) thus the bound lipids and posttranslational modifications (Musset B et al., 2010 JBC), (b) the detergent (DDM vs Fos-choline 12), (c) lipids (POPE/POPG vs POPC/POPE/POPS/SM/PI). Unfortunately, due to the time issue, we can not examine the effects of these factors on the results of liposome flux assays.

However, the most important point to our studies, is that nearly all hHv1 channel proteins are functional. In other words, nearly all hHv1 channels we studied are functionally available for voltage activation. Our studies were to examine the structural dynamics of the hHv1 channels at different gating states and uncover the changes induced by voltage and/or pH. Lower relative activities perhaps indicate that hHv1 channels in our smFRET studies have overall lower open probabilities or smaller single channel conductances upon activating voltage/pH. As a result, smaller changes in FRET distributions or fewer transitions from resting to activated states will be observed on less active hHv1 channels upon activating voltage/pH.

In the revised manuscript (Line 168-174), we have discussed the reason why the FRET occupancy change at strong resting voltage -85 mV and strong activating voltage 120 mV is only 10 ~15% for high or low FRET state (Figure 1C and E). One possible cause could be that even under 120 mV, the open probability of hHv1 channels is well below 1, partly indicated by our liposome flux data in comparison to those obtained by Lee et al.

– Since proton flux measurements take so long despite of using very high concentrations of valinomycin, a control with empty liposomes treated with the same reconstitution protocol without hHv1 should be included to demonstrate that it is the channel that produced the observed proton fluxes and that they are not originated from liposomes' proton leak. The controls included in figure S1 are insufficient since leaky liposomes will show the same results even in the absence of protein.

We appreciate the reviewer's suggestions. We have included the liposome flux assay with empty liposomes, which showed no proton leaks (Figure 1—figure supplement 1C).

– A proton flux assay of the mutant H140A in the wild-type hHv1 background must be included to assure the single mutation does not alter the hHv1 function.

Since the results of H140 were removed from the manuscript, we did not include H140A flux assay data. However, the liposome flux data do indicate that H140A is functional to mediate proton uptake.

– There is an inconsistency between fluxes shown in figure 4c and S2c. The former showed higher activity of mutant K169C-Q194C compared to wild-type, while the latest show wild-type has higher or equal activity than the mutant. A similar incongruency is evident between figure S1c and S2b in the wild-type channel fluxes. Representative flux assays of each mutant must be included in a supplementary figures section to show the constancy and reproducibility of the functional assay.

As shown by Figure 1—figure supplement 1C and 2B, liposome flux assays, even on WT proteins, do have small variations batch by batch. Therefore, for every batch of liposome flux assay data, WT liposome samples were always included as controls. The channel activities of all mutants were normalized against the WT of the same batch, so they can be combined in one figure. We have included the liposome flux assay traces in the revised manuscript (Figure 1—figure supplement 2B).

On the results mentioned by the reviewers, we performed the flux assays of the two mutants at different times, each included a WT control. As shown in Author response image 1, the flux assay curve of the K169C-Q194C is quite similar to these of WT in the same batch (right panel). For clarity, we did not include all the WT hHv1 curves of the same batch (from the left panel) in the figure and apologize for the mistake.

**Author response image 1. sa2fig1:** 

– Protein reconstitution protocols for the functional assay and smFRET measurements are different. Is the hHv1 function comparable between these protocols? The authors must demonstrate that the function of the proteoliposomes is similar when the protein is reconstituted using these two different protocols.

We have examined the functions of the hHv1 channel in liposomes at different protein/lipid ratios, using different detergent removal methods. For clarity, we did not include these new results in the revised manuscript.

As shown by Author response image 2, the detergent removal methods do not impact the function of hHv1 channels in liposomes (left panel). At the protein lipid ratio of 1:4,000, the slope of AMCA fluorescence quenching and the ratio of ACMA fluorescence quenching after valinomycin and CCCP are slightly decreased. According to the simulations of Lee et al. (JMB, 2010), these differences are the results of different channel distributions in liposomes at different protein lipid ratios, rather than the activities of hHv1 channels.

Dialysis was used to prepare smFRET liposomes, mainly for it is slower in detergent removal than BioBeads SM2. We did find that liposomes prepared by dialysis contain less bright fluorescent spots perhaps from aggregated hHv1 channels during reconstitution.

Reviewer #3:The gating of Hv1 protons channels is distinguished from related voltage-sensitive ion channels and phosphatases by an apparently unique sensitivity to changes in both membrane potential and the transmembrane pH gradient (ΔpH = pHo – pHi). The mechanism of ΔpH sensitivity in Hv1 is of widespread interest, but remains only partially understood. In particular, extracellular ionizable residues that are postulated to be required for sensing changes in pHo remain unidentified.Here the authors utilize a FRET approach in which recombinant, purified Hv1 channels carrying Cys mutations at pairs of extra- and intra-cellular residues are labeled with separate fluorophores and fluorescence changes (and the corresponding FRET ratios) are measured optically over time under conditions that are predicted to alter membrane potential and at various pHi. Although the development of a new optical assay to indirectly measure Hv1 channel activity represents a potentially important advance in the field, the data on which the authors base their main conclusions can be explained by alternative mechanisms that have not been ruled out. Furthermore, the authors' putative identification of a single mutation (H140A) that is purported to abolish pHo sensitivity is inconsistent with previous study showing that simultaneous mutation of both candidate extracellularly-exposed His residues in Hv1 (H140A-H193A) is insufficient to abolish pHo-dependent shifts in the apparent POPEN-V relation measured using voltage clamp electrophysiology 1.(The previous electrophysiology data is particularly problematic. Please clarify how you expect this to be reconciled.)The caveats to interpretation of the data presented in this manuscript are many, but several central concerns are outlined here:1. FRET changes measured here are not independently correlated with channel activity measured using electrophysiological methods, and it remains unclear what effects the double-Cys mutations (either before or after labeling) may have had on the voltage dependence and/or kinetics of channel gating. The reader must assume that mutant channels are WT-like, but a previously published survey of many Hv1 mutations clearly shows that altering channel structure can have major impacts on functional parameters 1.(Address the caveats in discussion..)

First, the mutations of many residues in hHv1 channels have been characterized previously (DeCoursey TE et al. 2016), including S98, C107, K125 and K169. Mutations of these residues only shift the threshold activating voltages to -7 (to Ala), -17 (to Ala), 19 (to Ala) and -13 mV (to Val), which are very close to ~5 mV of the WT channel. Deletion of C-terminal after I213 (including S219 and S224) also does not shift the threshold activating voltages significantly, thus we do not anticipate a dramatic change in voltage gating will cause by charge-neutral S219 and S224 mutations. More importantly, we choose -85 and 120 mV as potentials to stabilize hHv1 channels at resting and activating states, and no mutations mentioned above are likely to dramatically shift the resting potential below -85 mV or the activating voltage above 120 mV.

In Figure S2b, 2c, we have examined the function of fluorophore-labeled hHv1 cysteine mutants with liposome flux assays. The results indicated that except for K125C-K169C, all the hHv1 proteins, driven by ~±60 mV (i.e electrical potential generated by 150 mM K^+^ inside and 15 mM K^+^ outside, depending on channel orientations in liposomes), remain functional to conduct protons.

In addition, the smFRET data were cross-validated by multiple labeling sites, for example, K125C-S224C vs K169C-Q194C, which report the consistent voltage-dependent shift in FRET distributions. Therefore, it is very unlikely that the activating voltages of both mutants shift out of the voltage range between -85 mV and 120 mV.

2. The authors interpret FRET changes to selectively report movement of the S4 helix, but Cys residues are introduced into predicted loops between helical segments, and could also report local changes that occur independently of S4 helix movements that are thought to underlie voltage-dependent gating. There appears to be no independent experimental evidence that FRET changes actually report S4 movement. Of concern, only one mutant pair (K169C-Q194C) appears to exhibit a substantial voltage-dependent change in FRET ratio (5 other Cys mutant pairs exhibit little or no apparent voltage-dependent FRET changes), and data in Figure 1f showing this change contains undescribed error bars; neither is any statistical method for quantifying the magnitude of the voltage-dependent FRET change described in Methods.(This type of concern is understandable but the fact that you are measuring conformational distributions indicates that the channel S4 is detecting different conformations although it may be hard to predict the specific details of this conformation)

Our smFRET data reflect the conformational distributions and their changes induced by pHs and voltages. Mapping of smFRET data from multiple labeling sites onto the hHv1 channel structure agrees well on a structural model suggesting the upward movements of the S4 segment driven by activating voltage/pH.

In the revised manuscript, we simulated the accessible volumes of the fluorophores attaching at different labeling sites and then predicted the FRET changes when hHv1 channels transiting from the ‘resting’ to ‘activated’ conformations, provided by long time molecular dynamic simulations. The predicted FRET changes from 0.28 to 0.57 at the K125C-S224C labeling sites, from 0.72 to 0.45 at the K169C-Q194C labeling sites, match well with 2 out 3 FRET peaks identified by our smFRET studies (Line 264-275).

We have now described the error bars of all data presented and included statistical tests (unpaired two-tailed t-test) to objectively examine the voltage and pH-dependent changes in conformational distribution changes (i.e FRET state occupancies).

3. The data shown in Figure 4d are interpreted to mean that H140A mutation abolishes pHo sensitivity measured by pH-dependent quenching of a pH-sensitive dye (ACMA) within liposomes containing purified and reconstituted Hv1 channels (a similar method was previously described 2, but is evidently not cited in the References). The authors state that "on the H140A mutation background, the FRET distributions at the K169C-Q194C labeling sites do shift by voltage (Figure 4d)", which seems to mainly argue that this mutant channel remains voltage-dependent, and evidently does not directly address whether pHo sensitivity is altered. Furthermore, a previous study showed that H140A-H193A mutant Hv1 channels retain WT-like pHo sensitivity (-46 mV/pHo unit shift in VTHR, which is similar to the -38 mV/pHo unit shift measured for WT Hv1; see Table S1) 1.

The H140A/193A mutant still has the pHo sensitivities, like that of WT hHv1 channels.

Lowering of extracellular pH positively shifts the activation voltage of hHv1 channels. One possible explanation could be that H140A may negatively shift the activation voltage of hHv1 channels by stabilizing the high FRET conformations of the S4 segment, like the H168Q mutation. As a result, with H140A mutation, voltage-dependent changes in hHv1 open probability still can be observed within the voltage range from -85 mV to 120 mV.

To illustrate the explanation, we calculated the open probability of the hHv1 channels with the Boltzmann equation (as shown in Author response image 3), where the elementary charge z is ~3.

**Author response image 3. sa2fig3:** 

We assume the half activation voltage (V_mid_) of WT hHv1 channels is ~10 mV, without proton inhibition by

introducing N214R mutation, from -85 mV to 120 mV, the open probability (Po) changes from ~0 to 1. If the external pH drops by 3 pH units, without N214R mutation, it could shift V_mid_ to 140 mV. Then from -85 to 120 mV, the Po only changes from 0 to less than 0.1. However, if the H140A mutation shifts the V_mid_ by -30 mV (i.e 110 mV), the Po change could be from 0 to ~0.8. This could be one possible explanation on why we can not observe FRET changes without N214R mutation, but can with H140A mutation.

However, we decided to remove the results of the H140A mutation part, for its exact roles in hHv1 channels still need to be carefully characterized.

Recommendations for the authors:1. Determine whether mutations alter biophysical properties of expressed Hv1 channels.

As we have mentioned in our responses above, many mutations have been characterized previously by electrophysiological studies and no significant shifts in threshold activating voltage were found on these mutants. Our liposome flux assay data also showed that the hHv1 proteins for smFRET studies are functional to conduct protons at the voltage of ~±60 mV. Moreover, our smFRET studies were performed on the chemically modified (with fluorophores), monomeric hHv1 channels without posttranslational modifications (i.e obtained from *E. coli* expression system). Although we can not examine the biophysical properties of the expressed hHv1 channels in the revised manuscript for lacking the necessary electrophysiological equipment, the available electrophysiological data of others and the liposome flux data included in the manuscript support that conformational changes revealed by our smFRET studies are associated with voltage and pH gating (Line 114-117).

2. Establish that FRET changes necessarily report conformational rearrangements of the S4 helix, and not other motions that may be correlated with but not causal to voltage-dependent gating.

Please see the responses above, we have compared the FRET values we measured with those predicted from the structures from a recent simulation study using FRET positioning and screen software package and included the results in the revised manuscript (Line 264-275).

3. Address why FRET data are evidently contradictory to previously published electrophysiological data.

As we have explained, in response to concern #3, we have removed the results of H140A in the revised manuscript for its roles in hHv1 channel gating remain to be determined by mutational analysis and patch-clamp electrophysiology.

[Editors' note: further revisions were suggested prior to acceptance, as described below.]

The manuscript has been improved but there are some remaining issues that need to be addressed in the text, as outlined below:1. Overall, the authors basic observation that changes in membrane potential alter S4 position in a way that is consistent with the general consensus of voltage sensor function and previous reports of Hv1 specifically. Despite the potential of smFRET to reveal new insights into conformational rearrangements that occur during Hv1 channel gating, caveats to the interpretations of the data limit the value of the information in the current manuscript, and the work does not significantly advance knowledge in the field.Data in Figure 3 lead the authors to hypothesize that in H168Q, more negative voltage is required to move S4 into the resting (down) conformation than would be required for WT Hv1. A previous report1 shows that whereas both WT Hv1 and H168T manifest qualitatively similar positive shifts in the positions of their G-V curves as pHi is raised, the magnitude of the pH sensitive-shift in the position of the G-V curve is shallower (~20 mV/pHi unit) in H168T than in WT Hv1 (~40 mv/pHi unit). A positive shift in the G-V (i.e., at higher pHi) shows that channel closing occurs at less hyperpolarized potentials (or opening occurs at more depolarized voltages), which is opposite to the conclusion reached by the authors. One possible explanation is that the direction of proton currents at positive potentials is inward for H168T at high pHi all voltages between Vthr and the Nernst potential for H^+^ (EH^+^). Because the G-V is less shifted at acidic pHi in H168 mutants, there is a higher likelihood that the net current is inward, and if proton fluxes are oppositely directed in WT and H168Q channels, one might expect dramatically different effects on intra-liposomal pH (pHo) to result, as demonstrated by the authors for WT vs. N214R.The apparent effects of pHi acidification on S4 position is complicated by the possibility pHi is unknown under the experimental conditions used here.Please discuss these concerns in the discussion.

First, data in Figure 3 suggested that compared with these on the WT background, H168Q mutation enriches the FRET states representing activated conformations (i.e high FRET 0.9 population at K125C-S224C labeling sites and low FRET 0.24 population at the K169CQ194C labeling sites), therefore more negative voltage is required to stabilize the S4 segment at resting conformation.

As shown by the following two figures, (Figure 1F of WT hHv1 from Cherny et al., JGP 2015, and Figure 5G of H168T from Cherny et al., JGP 2018) at pHo 7, the G-V curve of WT at pHi 7 shows a threshold voltage of ~10 mV (black squares), while that of H168T shows a threshold voltage of -20 mV (blue dots). The data suggested that the H168T mutation shifts the G-V curve of hHv1 channels by -30 mV, which is consistent with our observations that on the H168Q mutation background, the occupancies of the FRET states representing activated conformations are higher than these on the WT background, at all pHi tested.

1. Cherny et al. Tryptophan 207 is crucial to the unique properties of the human voltage-gated proton channel, hHv1. J Gen Physiol, doi:10.1085/jgp.2015114562. Cherny et al. Histidine(168) is crucial for DeltapH-dependent gating of the human voltagegated proton channel, hHV1. J Gen Physiol, doi:10.1085/jgp.201711968 (2018).

Second, in the paper of Cherny et al. (JGP 2018), the data indicated that the voltagedependence of the H168T is less sensitive to pHi than the WT channel (i.e 20 mV/pHi vs 40 mV/pHi). But, the G-V curves of both WT and H168T channels are positively shifted with higher pHi, i.e the H168T or H168Q mutations do not completely abolish the sensitivity of hHv1 channels to pHi. However, we did not observe significant pHi-dependent changes in FRET distributions on the H168Q mutant. As discussed previously, the H168T mutation shifts the Vthreshold (probably also G-Vs) for -30 mV and decreases pH sensitivity to 20 mV per pHi. Thus increasing pHi by 2 units only shift the G-V curve for +40 mV in the H168T mutant, which is mostly canceled by the -30 mV shift caused by the H168T mutation. This could explain why at pHi 8.5, the FRET distribution of hHv1 channels with H168Q mutation is similar to that of on the WT background at pHi 6.5.

Third, on the WT background, the Vthreshold of hHv1 channels is 10 mV and the pH sensitivity is high (>40 mV per pHi). Thus the estimated Vthreshold at pHi 6.5, 7.5, 8.5 could be -30, 10, 70 mV, respectively. Since our smFRET data in Figure 3 were collected at 0 mV, this could explain the distinct FRET distributions at pHi 6.5 (strongly activated by 0 mV), in comparison with those at pHi 7.5 and pHi 8.5 (less activated by 0 mV), on the WT background.

Finally, if the H168Q mutant is activated at 0 mV at pHi 8.5, the proton efflux from liposomes (i.e pHo=pH7.5) following the proton chemical gradient will increase pHo, which could negatively shift the G-V curve. As a result, the FRET populations representing activated conformations could be enriched due to the pHo increase. This could also partially explain the FRET distributions at pHi 8.5 on the H168Q mutation background.

With N214R mutation, the proton flow into liposomes is almost completely abolished (i.e the outward H current), as shown by the liposome flux data of WT and N214R (Figure 1, figure supplement 2B). Thus at pHi 6.5, even hHv1 channels are activated, the proton flow into liposomes (i.e to decrease pHo) may not be significant enough to positively shift the G-V curve of hHv1 channels.

2. An alternative explanation for the data in Figure 3 not stated by the authors is that Hv1 H168Q channels enter a closed-state conformation in which nonetheless S4 remains in the activated (out) position, but this seems unlikely given that Hv1 channels do not appear to inactivate. The apparent paradox requires explanation and/or demonstration that membrane potentials and intra-liposomal pH remain intact during the experiments with H168Q; similar caveats apply to measurements of fluorescently-labeled WT (K125C-S224C and K169C-Q194C) channels. Unfortunately, it's not clear to me that the authors will be able to measure membrane potential and/or intra-liposomal pH (pHo) may not be within the authors av.Cherny, V. V., Morgan, D., Thomas, S., Smith, S. M. E. & DeCoursey, T. E. Histidine(168) is crucial for DeltapH-dependent gating of the human voltage-gated proton channel, hHV1. J Gen Physiol, doi:10.1085/jgp.201711968 (2018).In order to provide a nuanced perspective, please discuss the above limitations and its impact on your conclusions.

The K125C-S224C and K169C-Q194C channels on the WT background, as we have mentioned, can be complicated by pHo changes. As a result, the FRET distributions at positive activating voltages may not be necessarily reported as enrichments of activated conformations.

We do agree that without experimental measurements of liposome voltages and intraliposomal pHs, we should clarify these limitations in the discussion. We have revised our manuscript to reflect these limitations.

3. There are still a number of typos/mistakes in the revised version. Please have it proofread carefully if needed by an experienced colleague.

With our apologies, we have checked the manuscript very carefully to eliminate any typos and mistakes, with help from the experienced colleagues of our school.